# High-resolution distributions of $\Delta(O_2/Ar)$ on the northern slope of the
# South China Sea and estimates of net community production
Chuan Qin[1,2], Guiling Zhang[1,2,*], Wenjing Zheng[1], Yu Han[1,3], Sumei Liu[1,2]
1. Key Laboratory of Marine Chemistry Theory and Technology, Ministry of Education/Institute for
Advanced Ocean Study, Ocean University of China, 238 Songling Road, 266100 Qingdao, P. R.
China
2. Laboratory for Marine Ecology and Environmental Science, Qingdao National Laboratory for
Marine Science and Technology, Qingdao 266237, P. R. China
3. Hainan Tropical Ocean University, Sanya 572022, P. R. China
* Correspondence to: Guiling Zhang (guilingzhang@ouc.edu.cn)
## Abstract
Dissolved oxygen-to-argon ratio ($O_2/Ar$) in the oceanic mixed layer has been widely used to
estimate net community production (NCP), which is the difference between gross primary
production and community respiration and is a measure for the strength of the biological pump.
In order to obtain the high-resolution distribution of NCP and improve our understanding of its
regulating factors in the slope region of the Northern South China Sea (SCS), we conducted
continuous measurements of dissolved $O_2$, Ar, and $CO_2$ by membrane inlet mass spectrometry
(MIMS) during two cruises in October 2014 and June 2015. An overall autotrophic condition
was observed in the study region in both cruises with an average $\Delta(O_2/Ar)$ of 1.1 % ± 0.9 % in
October 2014 and 2.7 % ± 2.8 % in June 2015. NCP was on average 11.5 ± 8.7 mmol C m$^{-2}$ d$^{-1}$
in October 2014 and 11.6 ± 12.7 mmol C m$^{-2}$ d$^{-1}$ in June 2015. Correlations between dissolved
inorganic nitrogen (DIN), $\Delta(O_2/Ar)$, and NCP were observed in both cruises, indicating that
NCP is subject to the nitrogen limitation in the study region. In June 2015, we observed a rapid
response of the ecosystem to the episodic nutrient supply induced by eddies. Eddy-entrained
shelf water intrusion, which supplied large amounts of terrigenous nitrogen to the study region,
promoted NCP in the study region by potentially more than threefold. In addition, upwelling
brought large uncertainties to the estimation of NCP in the core region of the cold eddy (cyclone)
in June 2015. The deep euphotic depth in the SCS and the absence of correlation between NCP
and the average photosynthetically available radiation (PAR) in the mixed layer in the autumn

indicate that light availability may not be a significant limitation on NCP in the SCS. This study helps to understand the carbon cycle in the highly dynamic shelf system.

*Keywords*: $O_2/Ar$; Net community production; Nutrients; Eddy; Northern slope of South China Sea

## 1. Introduction

The oceanic carbon sequestration is partially regulated by the production and export process of biological organic carbon in the surface ocean. Net community production (NCP) corresponds to gross primary production (GPP) minus community respiration (CR) in the water (Lockwood et al., 2012) and is an important indicator of carbon export. At steady state, NCP is equivalent to the rate of organic carbon export, and is a measure for the strength of the biological pump (Lockwood et al., 2012). NCP effectively couples carbon cycle and oxygen ($O_2$) production through photosynthesis and respiration in the euphotic layer, thus many previous researches measured the mass balance of $O_2$ to quantify NCP (e.g., Emerson et al., 1991; Hendricks et al., 2004; Huang et al., 2012; Reuer et al., 2007). Argon (Ar), a biological inert gas, was commonly used to normalize the $O_2$ concentration in these researches. Based on the similar solubility properties of $O_2$ and Ar, oxygen-to-argon ratio ($O_2/Ar$) can remove the influences of physical processes (i.e., temperature and pressure change, bubble injection) on the mass balance of $O_2$ (Craig and Hayward, 1987). Dissolved $O_2/Ar$ has been developed as a proxy for NCP in a water mass (Kaiser et al., 2005). The biological production in the open oceans (i.e., Southern Ocean, Pacific, Arctic Ocean) has been inferred using the $O_2/Ar$ ratio to estimate NCP in numerous researches (e.g., Hamme et al., 2012; Lockwood et al., 2012; Ulfsbo et al., 2014; Shadwick et al., 2015; Stanley et al., 2010). During recent years, several high-resolution measurements of $O_2/Ar$ and NCP in coastal waters have been reported (Tortell et al., 2012; Tortell et al., 2014; Eveleth et al., 2017; Izett et al., 2018). Despite the coastal waters such as shelves and estuaries only accounting for 7 % of the global ocean surface area, they are known to

contribute to 15–30 % of the total oceanic primary production (Bi et al., 2013; Cai et
al., 2011) and play an important role in marine carbon cycle and production. However,
these regions still suffer from low resolution measurements that can't provide
representative high-resolution NCP data.
The South China Sea (SCS) is one of the largest marginal seas in the world with
complex ecological characteristics. River runoff from the Pearl and Mekong Rivers
introduces large amounts of dissolved nutrients into the SCS (Ning et al., 2004). Due
to the influence of seasonal monsoons, the surface circulation in the SCS changes from
a basin-scale cyclonic gyre in winter to an anticyclonic gyre in summer (Hu et al., 2000).
The surface water masses on the northern slope of SCS can be categorized into three
regimes: shelf water, offshore water (e.g., the intruded Kuroshio water), and the SCS
water (Feng, 1999; Li et al., 2018). The shelf water is mixed with fresh water from
rivers or coastal currents and thus usually has low salinity (S < 33) and low density (Uu
and Brankart, 1997; Su and Yuan, 2005; Cheng et al., 2014). Both offshore water and
SCS water originate from the Northern Pacific. Thus offshore water has similar
hydrographic characteristics of high temperature and high salinity as the Northern
Pacific water. But the SCS water has changed a lot in its hydrographic property because
of the mixing processes, heat exchange and precipitation during its long residence time
of about 40 years in the SCS (Feng et al., 1999; Li et al., 2018; Su and Yuan, 2005).
The distributions of phytoplankton and primary productivity of the SCS show great
temporal and spatial variation (Ning et al., 2004). Low chlorophyll a (Chl a) and
primary production are the significant characteristics of the SCS basin which is
considered an oligotrophic region, and macronutrients (i.e. nitrogen) are the main
limitations of phytoplankton growth and productivity (Ning et al., 2004; Lee Chen,
2005; Han et al., 2013). The excessive runoff form Pearl River can result in high N/P
(nitrogen/phosphorus) ratio of > 100, shifting the nutritive state from nitrogen
deficiency to phosphorus deficiency in the coastal region of SCS (Lee Chen and Chen,
2006). Dissolved iron is also a potential limitation on primary production, especially in
the high nutrient low chlorophyll (HNLC) regions (Cassar et al., 2011). But on the
northern slope of the SCS, the concentration of dissolved iron is high enough to support

the growth of phytoplankton in the surface water (Zhang et al., 2019). The northern

slope of the SCS is an important transition region between coastal area and the SCS

basin. In the summer, the shelf water intrusion is an important process changing the

nutritive state in the northern slope region of the SCS (He et al., 2016; Lee Chen and

Chen, 2006). But so far, the NCP enhancement caused by this process is still unknown.

 Previous studies about the organic carbon export in the SCS were mostly conducted

on particulate organic carbon (POC) flux (e.g., Bi et al., 2013; Cai et al., 2015; Chen et

al., 1998; Chen et al., 2008; Ma et al., 2008; Ma et al., 2011). Little research has been

conducted on NCP in the SCS to date. Chou et al. (2006) estimated NCP in the northern

SCS during the summertime to be 4.47 mmol C $m^{-2}$ $d^{-1}$ based on the time change rate

of dissolved inorganic carbon (DIC) in the mixed layer at the South East Asia Time-

Series Station (SEATS) from 2002 to 2004. Wang et al. (2014) used GPP and CR data

from a light/dark bottle incubation experiment to calculate NCP in the northern SCS

and obtained a range from −179.0 to 377.6 mmol $O_2$ $m^{-2}$ $d^{-1}$ (− 129.7 to 273.6 mmol C

$m^{-2}$ $d^{-1}$). Huang et al. (2018) estimated monthly NCP from July 2014 to July 2015 based

on in situ $O_2$ measurements on an Argo profiling float and reported the cumulative NCP

to be 0.29 mol C $m^{-2}$ $month^{-1}$ (9.67 mmol C $m^{-2}$ $d^{-1}$) during the northeast monsoon

period and 0.17 mol C $m^{-2}$ $month^{-1}$ (5.67 mmol C $m^{-2}$ $d^{-1}$) during the southwest

monsoon period in the SCS basin. However, most of these studies in the SCS were

constrained by methodological factors attributed to discrete sampling and cannot reveal

the rapid productivity response to highly dynamic environmental fluctuations of coastal

systems. Discrete sampling suffers from low spatial resolution, and cannot adequately

resolve variabilities caused by small-scale physical or biological processes in the

dynamic marine systems. In addition, each of the three methods for NCP estimate

mentioned above has its limitation. DIC-based NCP estimate is not suitable for the

coastal region, because instead of biological metabolism, the terrestrial runoff can be

the strongest factor influencing the DIC in the coastal system (Mathis et al., 2011). The

inavoidable difference between in situ circumstance and on-deck incubation condition

can introduce uncertainties to the NCP derived from light/dark bottle incubation

(Grande et al., 1989). Though Argo profiling float partly gets rid of the limitation of

discrete sampling, it's hard to control its movement in the study region. However, no high-resolution measurement of NCP has been reported for the SCS so far.

In this paper, we present high-resolution NCP estimates in the northern slope region of the SCS based on continuous shipboard dissolved $O_2/Ar$ measurement. We discuss the regulating factors of NCP based on ancillary measurements of other hydrographic parameters. Our high-resolution measurements caught the rapid response of the ecosystem to the episodic nutrient supply induced by eddies and help us to quantify the contribution of eddy-entrained shelf water intrusion to NCP in the summer cruise.

## 2. Methods

### 2.1 Continuous underway sampling and measurement

Continuous measurements of dissolved gases ($O_2$, Ar, and $CO_2$) were obtained using membrane inlet mass spectrometry (MIMS, HPR 40, Hiden Analytical, UK) (Tortell, 2005) onboard the *RV* 'Nanfeng' during two cruises in the northern slope region of the SCS (Figures 1a, 1b) from 13 to 23 October 2014 and from 13 to 29 June 2015. In addition, a cyclonic-anticyclonic eddy pair was observed in June 2015 (Figure 1c) and resulted in dramatic influences on the study region.

We developed a continuous shipboard measurement system of dissolved gases following the method described by Guéguen and Tortell (2008). Surface seawater was collected continuously using the ship's underway intake system (~5 m depth) and was divided into different lines for various underway scientific measurements. Seawater from the first line passed through a chamber at a flowrate of 2–3 L min$^{-1}$ to remove macroscopic bubbles and to avoid pressure bursts. A flow of ~220 mL min$^{-1}$ was continuously pumped from the chamber using a Masterflex Peristaltic Pump equipped with L/S® multichannel cartridge pump heads (Cole Parmer). In order to minimize the $O_2/Ar$ fluctuations due to temperature effects and water vapor pressure variations, the water samples flowed through a stainless steel coil (~6 m) with 0.6 mm wall thickness immersed in a water bath (Shanghai Bilon Instrument Co. Ltd, China) to achieve a constant temperature (~2 ℃ below the sea surface temperature), which avoided

temperature-induced supersaturation and subsequent bubble formation. Then the water
samples were introduced into a cuvette with a silicone membrane mounted on the inside.
The analyte gases were monitored by a Faraday cup detector in the vacuum chamber
after diffusion through the silicone membrane, and the signal intensities at the relevant
mass to charge (m/z) ratios (32, 40 and 44 for $O_2$, Ar and $CO_2$, respectively) were
recorded by MASsoft. Based on the continuous measurement of 50 L air-equilibrated
seawater, the long-term signal stability (measured as the coefficient of variation) over
12 h was 1.57 %, 3.75 % and 2.21 % for $O_2$, Ar and $CO_2$, respectively. Seawater from
the second line passed through a flow chamber, where an RBR Maestro (RBR, Canada)
was installed to continuously record temperature, salinity, dissolved oxygen (DO), and
Chl a. We didn't obtain continuous DO data in October 2014 because the DO sensor of
RBR broke down. A third line was used to drain the excess seawater. Underway
pipelines were flushed with freshwater or bleach every day, to avoid possible in-lines
biofouling. The data from the underway transects were exported to spreadsheets and
compiled into 5 min averages, and the comparisons of the gas data with other
hydrographic variables were based on the UTC time recorded for each measurement.
The $O_2$/Ar ratio measurements were calibrated with air-equilibrated seawater samples
at about 6–8 h intervals to monitor instrument drift and calculate $\Delta(O_2/Ar)$. These air-
equilibrated seawater samples were prefiltered (0.22 μm) and bubbled with ambient air
for at least 24 h to reach equilibrium at sea surface temperature (Guéguen and Tortell,
2008). For calibration, 800 mL of air-equilibrated seawater sample was transferred into
glass bottles and immediately drawn into the cuvette, where the first 200 mL of the
sample was used to flush the cuvette and pipelines. After 3 min recirculation of the
sample, the average signal intensity was obtained to calculate $O_2$/Ar. During the course
of measurements, flow rate and the temperature of water bath were both kept the same
as the underway measurements. The precision of MIMS-measured $O_2$/Ar was 0.22 %,
based on analyses of 20 duplicate samples in the laboratory test, which is comparable
to previous studies and sufficient to detect biologically driven gas fluctuations in
seawater (Tortell, 2005).
The instrumental $CO_2$ ion current was calibrated at about 12–24 h intervals using
equilibrated seawater standards as per Guéguen and Tortell (2008) during the survey in
June 2015. Prefiltered seawater (0.22 μm) was gently bubbled with dry $CO_2$ standards
(200, 400, and 800 ppm, provided by the Chinese National Institute of Metrology) at in
situ temperature. After 2 days of equilibrium, these standards were analyzed by MIMS
following the same procedure for measuring air-equilibrated seawater samples to obtain
a calibration curve between $CO_2$ signal intensity and mole fraction. The reproducibility
of these measurements was better than 5 % within 15 days. Then we used the empirical
equations reported by Takahashi et al. (2009) to convert the $CO_2$ mole fraction derived
from the calibration curve to the in situ partial pressure of $CO_2$ ($p$$CO_2$).
Chlorophyll-a (Chl a) data from the RBR sensor were linear calibrated against
extracted Chl a measurements of discrete seawater samples taken from the same
seawater outlet as for MIMS measurements. Samples were filtered through
polycarbonate filters (0.22 μm). The filter membranes were then packed with pre-
sterilized aluminum foil and stored in a freezer (−20 °C) until extraction by acetone and
analysis using a fluorimetric method (F-4500, HITACHI, Japan) described by Parsons
(1984). The mean residual of this calibration was $0.00 \pm 0.07$ μg $L^{-1}$.

**2.2 Estimation of NCP based on $O_2$/Ar measurements**
NCP in the mixed layer was estimated by the $O_2$/Ar mass balance from continuous
measurements. Due to similar physical properties of $O_2$ and Ar, $\Delta(O_2/Ar)$ is used as a
proxy of the biological $O_2$ supersaturation and is defined as (Craig and Hayward, 1987):
$$\Delta(O_2/Ar) = \frac{([O_2]/[Ar])}{([O_2]/[Ar])_{eq}} - 1$$

where $[O_2]/[Ar]$ is the measured dissolved $O_2$/Ar ratio of the mixed layer and
$([O_2]/[Ar])_{eq}$ is the measured dissolved $O_2$/Ar ratio of the air-equilibrated seawater
samples. $\Delta(O_2/Ar)$ is the percent deviation of the measured $O_2$/Ar ratio from the
equilibrium. Assuming a steady state and negligible physical supply, NCP is the air–
sea biological $O_2$ flux and can be estimated as (Reuer et al., 2007):
$$NCP \ (mmol \ C \ m^{-2} \ d^{-1}) \approx k_{O_2} \cdot [O_2]_{sat} \cdot \Delta(O_2/Ar) \cdot r_{C:O_2} \cdot \rho$$

where $k_{O_2}$ is the weighted gas transfer velocity of $O_2$ (m $d^{-1}$); $[O_2]_{sat}$ denotes the
saturation concentration of dissolved $O_2$ (μmol $kg^{-1}$) in the mixed layer, which is
calculated based on temperature and salinity (Weiss, 1970); $r_{C:O_2}$ is the photosynthetic
quotient of C and $O_2$ and was reported as 1:1.38 in the SCS (Jiang et al., 2011); ρ is
seawater density in units of kg $m^{-3}$ (Millero and Poisson, 1981). We estimated $k_{O_2}$
using the European Centre for Medium-Range Weather Forecasts (ECWMF) wind-
speed reanalysis data product with a 0.25° × 0.25° grid (https://www.ecmwf.int), the
parameterization by Wanninkhof (1992), and the gas exchange weighting algorithm by
Teeter et al. (2018). Teeter et al., (2018) pointed out that modern $O_2$/Ar method does
not strongly rely on the steady state assumption. When this assumption is violated, our
estimate does not represent the actual daily NCP but rather an estimate of NCP
weighted over the residence time of $O_2$ in the mixed layer and along the path of the
water parcel during that period. Thus the residence time of $O_2$ in the mixed layer is an
important implication of the weighted timescale of NCP before the measurement of
$O_2$/Ar. The residence time of $O_2$ (τ, d) in the mixed layer is estimated as the ratio of
mixed layer depth (MLD, m) to the gas transfer velocity of $O_2$ ( $k_{O_2}$ , m $d^{-1}$) (Jonsson
et al., 2013).

**2.3 Ancillary measurements and calculations**

Surface water samples for the nutrient analysis were collected from Niskin bottles
mounted on the CTD, where the samples were filtered through acid-cleaned acetate
cellulose filters (pore size: 0.4 μm). The filtrates were poisoned by $HgCl_2$ and stored in
the dark at 4 °C. In the laboratory, the nutrients were determined photometrically by an
auto-analyzer (QuAAtro, SEAL Analytical, Germany) with a precision better than 3 %.
MLD was defined by the $\Delta\sigma_t = 0.125$ kg $m^{-3}$ criterion (Monterey and Levitus, 1997).
The subsurface chlorophyll maximum layer (SCML) was observed using the
fluorescence sensor mounted on the CTD. SCML usually occurs at the bottom of
euphotic layer (Hanson et al., 2007; Liao et al., 2018; Teira et al., 2005). Because no
PAR (Photosynthetically Available Radiation) profile data were obtained in two cruises,
we decided to regard the depth of SCML as the euphotic depth ($Z_{eu}$). Both MLD and
$Z_{eu}$ were calculated at each station where the vertical CTD casts were made. The MLDs

for underway data between CTD stations was calculated using linear interpolation based on the distance between the underway points and nearest CTD stations. We matched the underway data to each CTD location using a combination of latitude/longitude threshold (latitude/longitude of CTD station ± 0.05°) and time threshold (end/start of stationary time ± 1 h), then took the averages of these underway data for further analysis with discrete nutrient concentrations.

The daily satellite chlorophyll data were obtained from the E.U. Copernicus Marine Service Information website (https://resources.marine.copernicus.eu). The product we used was provided by ACRI–ST company (Sophia Antipolis, France), with a space-time interpolation (the "Cloud Free"). The M_Map package for Matlab was applied to output satellite chlorophyll images (Pawlowicz, 2020). Daily and 8-day PAR data collected by MODIS-Aqua sensor were obtained from NASA's ocean color website (https://oceancolor.gsfc.nasa.gov/l3). The spatial resolution of both satellite products is 4 km, and we match the satellite PAR with CTD location by choosing the closest PAR data point to the CTD location. A light attenuation coefficient ($K_d$, $m^{-1}$) was used to estimate the average PAR in the mixed layer (Kirk 1994; Jerlov 1976):

$$K_d = \frac{4.605}{Z_{eu}}$$

## 3. Results and Discussion

### 3.1 Distributions of hydrographic parameters and gases

The distributions of temperature, salinity, Chl a, and $\Delta(O_2/Ar)$ during the autumn cruise (October 2014) are shown in Figure 2. Sea surface temperature (SST) ranged from 26.96 °C to 28.53 °C with an average of 27.82 ± 0.33 °C. Sea surface salinity (SSS) ranged from 33.28 to 34.11 with the low values occurring in the southeast of the region. Chl a concentration ranged from 0.01 to 0.71 µg $L^{-1}$ and was in an average of 0.18 ± 0.13 µg $L^{-1}$, which was comparable to the 11-year mean value (~ 0.2 mg $m^{-3}$) in the same region in October reported by Liu et al. (2014). $\Delta(O_2/Ar)$ values were in the range of −2.9–4.9 % (avg. 1.1 % ± 0.9 %) and slightly oversaturated in most areas (Figure 2d). Please note that all averages we have published in this paper are reported in the

format of *mean ± standard deviation*.
In June 2015, SST ranged from 29.28 °C to 32.24 °C and was in an average of 30.88
± 0.59 °C (Figure 3a). SSS ranged from 30.81 to 34.16. Transect 3 was significantly
characterized by low salinity (Figure 3b). He et al (2016) reported that this phenomenon
was influenced by the eddy-entrained Pearl River plume (shelf water) injected into the
SCS. Chl a varied in a range of 0.09–0.58 $\mu g\ L^{-1}$ in the study region. Under the influence
of this eddy-entrained shelf water, Chl a values higher than 0.30 $\mu g\ L^{-1}$ were observed
along Transect 3 (Figure 3c). In contrast, Chl a was in the range of 0.09–0.18 $\mu g\ L^{-1}$
along Transect 1 and 2. It was obvious that DO was much higher in the east side than
the west side in the study region (Figure 3d). $\Delta(O_2/Ar)$ ranged from −3.9–13.6 %. Most
of the $\Delta(O_2/Ar)$ values were positive in the study region (avg. 2.7 % ± 2.8 %), whereas
the negative values were concentrated along Transect 4 (Figure 3f). $\Delta(O_2/Ar)$ along
Transect 3 was in an average of 7.2 % ± 2.6 %, significantly higher than that of other
transects (Figure 3f). $pCO_2$ exhibited a high degree of spatial and temporal variability
and the high values mostly occurred on the west side of the study region (Figure 3e).
Resulting from the considerable low $pCO_2$ in Transect 3, the average $pCO_2$ (323 ± 93
µatm) in the study region was lower than those reported previously, i.e., 350–370 µatm
by Zhai et al (2009) and 340–350 µatm by Rehder and Sues (2001). Due to the influence
of the shelf water, the average $pCO_2$ in Transect 3 was 222 ± 33 µatm, with a range of
144–321 µatm. In the summer, shelf water mixed with Pearl River plume is the most
important factor influencing $pCO_2$ in the coastal and shelf region of the northern SCS,
which can result in the $pCO_2$ values as low as 150 µatm (Li et al., 2020). Here we apply
an average atmospheric $pCO_2$ of 382 µatm that observed in July 2015 in the northern
SCS (Li et al., 2020) to calculate the $pCO_2$ difference ($\Delta pCO_2$) between the surface
water and the atmosphere. $\Delta pCO_2$ ranged from −238 to −61 µatm along Transect 3,
indicative of a strong $CO_2$ sink.

**3.2 Mixed layer depth, euphotic depth and residence time of O$_2$ in the mixed layer**
MLD, euphotic depth ($Z_{eu}$) and the residence time of $O_2$ ($\tau$) in the mixed layer at CTD
stations of two cruises are shown in Table 1 and 2. In autumn 2014, MLD ranged from

27 to 81 m, with an average of $55 \pm 15$ m (Table 1). The average $Z_{eu}$ was $74 \pm 12$ m, approximately 20 m deeper than MLD (Table 1). The residence time of $O_2$ in the mixed layer ranged from 3 to 13 d (Table 1), comparable to a range of 1–2 weeks reported by previous studies (Izett et al., 2018; Manning et al., 2017). The average residence time of $O_2$ was $9 \pm 3$ d, indicating that our estimate generally quantified NCP over 9 days prior to the underway observation of $O_2/Ar$ during this cruise.

The average MLD in June 2015 was just $18 \pm 6$ m (Table 2). Significant shallow MLD occurred at two stations (J-10, J-11) located in Transect 3 (Table 2, Figure S1f). The low-salinity shelf water intrusion is the main cause of this shallow MLD of 8 m. The average $Z_{eu}$ was $58 \pm 18$ m, approximately 40 m deeper than MLD (Table 2). The residence time of $O_2$ in the mixed layer ranged from 2 to 12 d (Table 2), indicating a fast gas exchange in some stations. In addition, we also observed relatively obvious subsurface $O_2$ maxima in Transect 1 and 2 in summer 2015. But this phenomenon didn't exist in autumn 2014.

In both cruises, $Z_{eu}$ was observed obviously deeper than MLD. This result partly suggests that light availability may not be a limitation of NCP in the northern slope of SCS. Especially in the summer, $Z_{eu}$ extended to 2–7 times of MLD (Table 2), ensuring sufficient illumination in the mixed layer. But in the autumn when the thickness of mixed layer accounts for about 74 % of euphotic layer, the average light intensity in the mixed layer might be influenced by the exponentially light attenuation along depth.

**3.3 NCP in autumn and summer**

In October 2014, NCP in the northern slope of the SCS ranged from $-29.2$ to 42.7 mmol C m$^{-2}$ d$^{-1}$ (avg. $11.5 \pm 8.7$ mmol C m$^{-2}$ d$^{-1}$) and most of the region was net autotrophic (Figure 4a). The estimated NCP based on the $O_2/Ar$ values measured in this cruise is about 34 % of the net primary production rates of 34.3 mmol C m$^{-2}$ d$^{-1}$ measured by $^{14}$C bottle incubation (Sun X., personal communication), which was in agreement with previous research (Quay et al., 2010).

The average NCP in the study region was $11.6 \pm 12.7$ mmol C m$^{-2}$ d$^{-1}$ with a range of $-27.6$–61.4 mmol C m$^{-2}$ d$^{-1}$ in June 2015. A high NCP level was observed along

Transect 3 (Figure 4b). Eddy-entrained shelf water brought a large amount of
terrigenous nutrients from the shelf to the slope region along Transect 3 (He et al., 2016).
The average nitrate ($NO_3^-$) and nitrite ($NO_2^-$) concentrations in the surface water of
Transect 3 were $2.31 \pm 0.70$ µmol $L^{-1}$ and $0.04 \pm 0.01$ µmol $L^{-1}$ respectively (Figure
S1a, S1b); both values were much higher than those found in the other three transects
where $NO_3^-$ was in a range of < 0.03–0.69 µmol $L^{-1}$ and $NO_2^-$ was mostly below the
detection limit. Li et al. (2018) reported that the entire Transect 3 and part of Transect
4 were dominated by shelf water at the surface and we estimated NCP over these regions
where salinity lower than 33 as $23.8 \pm 10.7$ mmol C $m^{-2}$ $d^{-1}$ on average. We also
observed a warm eddy (anti-cyclone) covering most stations in Transects 1 and 2
(Figure 1b, 1c) during our survey in June 2015 (Chen et al., 2016). Anti-cyclonic eddies
can cause downwelling, deepening of the thermocline, and blocking of the supply of
nutrients from the deeper water (Ning et al., 2008; Shi et al., 2014). Consequently, a
warm eddy is expected to result in an oligotrophic condition in the surface water
associated with low Chl a concentrations and low production (Ning et al., 2004). As a
result, in the summer of 2015, the observed $NO_2^-$, $NO_3^-$, and $PO_4^{3-}$ (phosphate)
concentrations were almost below the detection limit in Transects 1 and 2 (Figure S1a,
S1b, S1d). NCP in Transect 1 and 2 was at a very low level (avg. $2.8 \pm 2.7$ mmol C $m^{-2}$
$d^{-1}$). Because of the significant high values of NCP over the regions with shelf water
intrusion, our NCP result in the summer of 2015 is averagely higher than the previous
values of 4.47 mmol C $m^{-2}$ $d^{-1}$ and 0.17 mol C $m^{-2}$ $month^{-1}$ (5.67 mmol C $m^{-2}$ $d^{-1}$)
based on DIC budget and Argo-$O_2$ respectively in the SCS (Chou et al., 2006; Huang
et al., 2018). However, NCP estimates based on both methods mentioned above suffer
from poor temporal and spatial coverage and do not allow for revealing rapid changes
in shelf systems. In contrast, continuous measurements of $O_2$/Ar allow us to capture
rapid variations in NCP along Transect 3 and resolve short-term productivity responses
to environmental fluctuations.

**3.4 Distribution of various parameters along representative transects**
We chose Transect 5 (Figure 1a) observed in October 2014 and Transect 4 (Figure 1b)
observed in June 2015 to show the distribution of various parameters.
The distribution of Chl a, $\Delta(O_2/Ar)$, and NCP showed similar trend along Transect 5
in October 2014 (Figure 5). There was a trough of temperature, showing a maximum
drawdown of ~ 0.6 °C compared to the average temperature in the study region (Figure
5a). But the temperature fluctuations shown here are too small to reflect a significant
upwelling that can easily cause ~ 2 °C drawdown of temperature in the upper layer (Lin
et al., 2013; Manning et al., 2017; Ning et al., 2004). A spike of Chl a occurred between
115.6°E and 115.7°E and was coincident with the peaks of $\Delta(O_2/Ar)$ and NCP (Figure
5b, 5c). The highest surface concentration of ammonium ($NH_4^+$) of 0.35 μmol $L^{-1}$ was
also observed between 115.6°E and 115.7°E in this transect and was predominantly
higher than the concentrations (0.07–0.17 μmol $L^{-1}$) in the other regions of this cruise
(Figure 5c, S2b). Because no significant obduction processes (i.e., upwelling,
entrainment, and diapycnal mixing) were reported in this region, the most likely source
of this abundant $NH_4^+$ was in situ regeneration such as the excretion of zooplankton and
the bacterial decomposition of organic matter (La Roche, 1983; Clark et al., 2008).
Theoretically, $NH_4^+$, an important nitrogen source of phytoplankton growth, can be
quickly utilized by phytoplankton, and contributes to primary production (Dugdale and
Goering, 1967; Tamminen, 1982). However, we only got nutrient data at two CTD
stations in this transect, thus the result we obtained here just indicated that high NCP
occurred at the station with relatively high $NH_4^+$ concentration, but couldn't be a strong
evidence that $NH_4^+$ was the main factor influencing NCP in this transect.
A similar distribution pattern of Chl a, NCP, and $\Delta(O_2/Ar)$ was observed along
Transect 4 in June 2015, whereas $p$CO_2 showed the opposite trend for these three
parameters (Figure 6b, 6c). Low salinity (lower than 33) existed at both southern and
northern ends of this transect (Figure 6a). The concentration of dissolved inorganic
nitrogen (DIN, $NO_3^- + NO_2^- + NH_4^+$) in the surface water was 0.81 μmol $L^{-1}$ and 0.27
μmol $L^{-1}$ at the southern and northern end respectively, which was higher than the
concentrations in other stations of this transect (Figure 6c). These results indicate that
shelf water is imported at the northern and southern ends of this transect, along with
higher levels of Chl a and NCP (Figure 6c). A sharp drop in the temperature and an
increase in salinity occurred from 19.7°N to 19.8°N and from 21°N to 20.7°N (Figure
6a), manifesting an upwelling over this area together with dramatic spikes in $p\text{CO}_2$ and
associated decrease in $\Delta(\text{O}_2/\text{Ar})$ (Nemcek et al., 2008) (Figure 6b). Most regions of
Transect 4 were dominated by upwelling and showed negative sea level height anomaly
(Chen et al., 2016; He et al., 2016). A localized cold eddy was considered the cause of
this upwelling (Figure 1c), resulting in a maximum temperature drawdown of ~1.6 °C
in the mixed layer.
Vertical mixing is considered the largest source of error in $\text{O}_2/\text{Ar}$-based NCP estimates
because the upwelled subsurface water with different $\text{O}_2/\text{Ar}$ signatures can produce
either an overestimation or an underestimation of NCP in the mixed layer (Cassar et al.,
2014; Izett et al., 2018). Former researches usually ignored the underestimated negative
NCP that caused by vertical mixing (Giesbrecht et al., 2012; Reuer et al., 2007; Stanley
et al., 2010). Cassar et al. (2014) presented a $\text{N}_2\text{O}$-based correction method of $\text{O}_2/\text{Ar}$
and NCP for vertical mixing. Although this method has been successfully adopted by
Izett et al. (2018) in the Subarctic Northeast Pacific, it is not suitable for our study
region. This is because it is basically applicable in the areas where the depths of
euphotic zone and mixed layer are similar, and this method is not suitable for
oligotrophic regions (Cassar et al., 2014). The SCS is recognized as an oligotrophic
region and the depth of the euphotic zone can be 2–7 times that of the mixed layer in
our study region in the summer. In addition, in the region (e.g. the SCS basin) where
subsurface oxygen maximum exists, the applicability of $\text{N}_2\text{O}$-based correction method
is limited (Izett et al., 2018). In Transect 4, the regions with negative NCP and the
regions with salinity higher than 33.5 and temperature lower than 30 °C are defined as
influenced by upwelling. If we neglect these regions in Transect 4, the average NCP in
June 2015 can slightly raise to $12.4 \pm 12.3$ mmol C m$^{-2}$ d$^{-1}$. If we also remove the
influence of shelf water intrusion by neglecting the regions with salinity lower than 33,
the average NCP can sharply decrease to $5.0 \pm 6.2$ mmol C m$^{-2}$ d$^{-1}$, which was similar
to the results of 4.47 mmol C m$^{-2}$ d$^{-1}$ and 0.17 mol C m$^{-2}$ month$^{-1}$ (5.67 mmol C m$^{-2}$
d$^{-1}$) reported in previous researches in the same season (Chou et al., 2006; Huang et al.,
2018). Here we regard $5.0 \pm 6.2$ mmol C m$^{-2}$ d$^{-1}$ as the background value of NCP in the
study region. Since an average NCP of $23.8 \pm 10.7$ mmol C $m^{-2}$ $d^{-1}$ was observed over
the regions with salinity lower than 33, we can conclude that the summer shelf water
intrusion significantly promoted NCP by potentially more than threefold in June 2015.

**3.5 Factors influencing NCP in the SCS**

The SCS is an oligotrophic region with low biomass and primary production (Lee Chen,
2005; Ning et al., 2004). Previous research has shown that the nutrient, especially
nitrogen and phosphorus, is the most important factor controlling and limiting the
phytoplankton biomass and primary production in the SCS (Ning et al., 2004; Lee Chen,
2005; Lee Chen and Chen, 2006; Han et al., 2013). After neglecting the two CTD
stations (J-14, J-15) with negative NCP influenced by upwelling in June 2015, we
performed a principal component analysis (PCA) to determine the dominant factors
influencing NCP in both cruises. In October 2014, DIN (0.741), $\Delta(O_2/Ar)$ (0.858), and
NCP (0.979) were significantly loaded on Factor 1, indicating a potential relationship
among these three variables (Figure 7a, Table S1b). The correlation coefficient between
DIN and NCP was 0.706 ($p < 0.01$, Table S1a), which was significantly higher than the
coefficient between NCP and the other variables, except for $\Delta(O_2/Ar)$ and temperature;
this indicated that DIN was an important factor influencing NCP in this cruise. Another
two nutrients – dissolved silicate (DSi, $SiO_3^{2-}$) and dissolved inorganic phosphorus
(DIP, $PO_4^{3-}$) – had no correlations ($p > 0.05$) with NCP (Table S1a). In June 2015,
Factor 1 showed a strong loading by DIN (0.876), Chl a (0.950), DO (0.927), $\Delta(O_2/Ar)$
(0.902), and NCP (0.909), whereas salinity ($-0.936$) and $pCO_2$ ($-0.908$) were
negatively loaded on Factor 1 (Figure 7b, Table S2b). The injection of low salinity shelf
water appeared to have a strong effect on the study region because significant negative
correlations were observed between salinity and DIN, Chl a, $\Delta(O_2/Ar)$, and NCP (Table
S2a). DIN had strong correlations with NCP, $\Delta(O_2/Ar)$, and Chl a, with the correlation
coefficients of 0.747, 0.910, and 0.754, respectively (Table S2a), indicating that DIN
was the dominant factor controlling the growth of phytoplankton and primary
production in this cruise. DSi (0.582) and DIP ($-0.601$) were both moderately loaded
on Factor 2 (Figure 7b, Table S2b) and had no correlations with NCP ($p > 0.05$, Table

S2a). These results suggested the key role of nitrogen in regulating $\Delta(O_2/Ar)$, NCP, and

phytoplankton biomass in the SCS. The supply of nitrogen may stimulate the growth of

phytoplankton in the SCS and nitrogen is an important participant in photosynthesis

and a basic element that contributes to the increase in primary production (Dugdale and

Goering, 1967; Lee Chen, 2005; Lee Chen and Chen, 2006; Han et al., 2013).

Coupled with biochemical variations, physical processes also play important roles in

the slope region of the SCS by transporting abundant nutrient-rich shelf water into the

SCS and bringing deep water to the surface by enhancing water mixing (Chen and Tang,

2012; Ning et al., 2004; Pan et al., 2012). The surface waters in the slope region of the

northern SCS are primarily composed of waters originating from SCS water, Kuroshio

water, and shelf water (Li et al., 2018). In the summer, the shelf water exists where the

potential density anomaly is lower than 20.5 kg m$^{-3}$ (Li et al., 2018). In the autumn,

there is a weak offshore transport of the shelf water in the SCS and the salinity of the

water mixed with the shelf water is usually lower than 33 (Fan et al., 1988; Uu and

Brankart, 1997; Su and Yuan, 2005). In October 2014, the observed surface salinity was

in a range of 33.28 to 34.11; thus the surface waters were mainly derived from mixing

of the Kuroshio water and the SCS water. In the summer of 2015, a cyclonic-

anticyclonic eddy pair was observed in the study region (Figure 1c). Low-salinity shelf

water mixed with the intruding river plume from the Pearl River in the upper 50 m and

was transported to the slope and basin along the intersection of the two eddies (Chen et

al., 2016; He et al., 2016; Li et al., 2018). In both seasons, the surface waters in the

study region were generally found to be nitrogen deficient, with $NO_2^-$ at < 0.01–0.04

μmol L$^{-1}$ (Figure S2a, S1b), $NO_3^-$ at < 0.03–2.82 μmol L$^{-1}$ (Figure S1a), and $NH_4^+$ at

0.04–0.35 μmol L$^{-1}$ (Figure S2b, S1c). The concentrations of $NO_2^-$ and $NO_3^-$ were

below the detection limit at almost 80% of the sampling stations during both cruises.

Due to the injection of shelf water with low salinity and abundant terrestrial nutrients,

significant high concentrations of $NO_3^-$ and $NO_2^-$ were observed along Transect 3 in

June 2015 (Figure S1a, S1b) where the shelf water was intruded by eddies (Chen et al.,

2016; He et al., 2016). Such transport processes from the inner shelf to the slope region

have a profound influence on nutrient dynamics and biological productions (He et al.,

2016). The water that was influenced by shelf water with a potential density anomaly lower than 20.25 kg m$^{-3}$ and salinity lower than 33 had high concentrations of DIN (Figure 8a). At the 6 stations (in the red circle of Figure 8a) that were intruded by shelf water and characterized with surface salinity lower than 33, we obtained an average surface DIN concentration of $1.82 \pm 1.16$ (0.27–3.01) µmol L$^{-1}$, which was significantly higher than the mean of $0.10 \pm 0.03$ (0.04–0.16) µmol L$^{-1}$ at other stations (independent samples t-test, $p < 0.01$). After neglecting the two stations (J-14, J-15) influenced by upwelling, a strong correlation between NCP and DIN was observed in the cruise of June 2015 (r = 0.747, $p < 0.01$), with higher NCP (avg. $15.4 \pm 4.5$ mmol C m$^{-2}$ d$^{-1}$) occurred at the stations where shelf water intruded, consistent with the DIN concentration higher than 0.27 µmol L$^{-1}$ (Figure 8b). At other stations without the influence of shelf water, the average NCP was just $2.3 \pm 1.7$ mmol C m$^{-2}$ d$^{-1}$. These results furtherly suggest that the supply of DIN from shelf water can greatly stimulate the primary production at these stations, resulting in the NCP increase of nearly 7 times compared to other stations.

The correlations between NCP and sea surface temperature and salinity also support the influence of physical forcing on NCP. In June 2015, we obtained a strong negative correlation between NCP and salinity (Figure 9d). NCP significantly increased in the water with salinity lower than 33 (Figure 9d). Temperature had weak correlations with NCP (Figure 9c), and the negative NCP values were concentrated in the water with temperatures below 30.5 °C and salinity values over 33.5 (Figure 9c, 9d). This surface water was mostly observed along Transect 4 where vertical mixing caused by a cold eddy brought deep water to the surface. The undersaturated $\Delta(O_2/Ar)$ entrained by deep water caused the negative NCP estimates at the surface, resulting in a considerable underestimation of NCP. Unlike in June 2015, all the correlations were very weak between NCP and temperature, salinity in October 2014 (Figure 9a, 9b). The Kuroshio water and the SCS water had similar hydrological characteristics and their mixing in October 2014 may not have resulted in significant changes in the hydrological characteristics of the surface water.

The nutrient concentrations and hydrographic characteristics we observed just reflect

the marine environment at the moment of sampling, partly contradicting our estimates
that quantified NCP over a period prior to the observation. Especially for the regions
with significant influence of shelf water in June 2015, tracking the history of shelf water
intrusion is important. We used daily satellite chlorophyll data to monitor the intrusion
of shelf water and roughly set satellite-chlorophyll $\geq \sim 0.2$ μg L$^{-1}$ as the criterion of
shelf water (Figure 10). On 10 June 2015, shelf water began to influence the northern
end (J-9) of Transect 3 and most part of Transect 4, then it extended to the southern end
of Transect 3 and Transect 4 where J-12 and J-13 located on 13 June (Figure 1b, 10).
Till 25 June when we finished the observation of Transect 4, the entire Transect 3 (J-9
to 12) as well as J-13 and J-16 had kept been dominated by shelf water for more than
10 days (Figure 1b, 10). We concluded these findings in Table 3, along with the
residence time ($\tau$) of O$_2$ in the mixed layer and the difference ($\Delta$day) between the date
of observation and the start date of shelf water intrusion at the stations with surface
salinity lower than 33. $\Delta$day can represent the duration of the shelf water intrusion at
each station before our observation. The residence time of O$_2$ in the mixed layer at most
stations listed in Table 3 is shorter than or equivalent to $\Delta$day. This result suggests that
our estimate has appropriately integrated the NCP during the period of shelf water
intrusion, which can effectively reflect the influence of shelf water on productive state
on the northern slope of the SCS in the summer.
The amount of light may also play a role in the extent of primary production. The
MLD is considered a driver of light availability in the mixed layer (Cassar et al., 2011;
Hahm et al., 2014). The euphotic layer was averagely 40 m thicker than the mixed layer
in the study region during the summer cruise, thus it's not very significant to discuss
the light limitation in June 2015. We conducted an analysis of light availability based
on daily satellite-PAR data and NCP in October 2014. To minimize the influence of
DIN concentrations, we selected 9 stations where surface DIN concentration in the
range of 0.10—0.17 μmol L$^{-1}$. The average surface PAR (mol m$^{-2}$ d$^{-1}$) at each station
was integrated over the residence time of O$_2$ before our observation. Then an average
PAR in the mixed layer was calculated based on K$_d$. At the selected stations, the surface
PAR varies over a range of 38.6—42.2 mol m$^{-2}$ d$^{-1}$, while the average PAR in the mixed

layer (ML PAR) ranged from 8.7 to 13.3 mol m$^{-2}$ d$^{-1}$ (Table 4). There's no significant correlation between the average PAR and NCP in the mixed layer (Table 4), partly suggesting that light intensity may not be a factor on NCP in the autumn. Light availability in the northern slope region of SCS is enough to support the primary production of phytoplankton.

## 4 Conclusion

The distribution of $\Delta(O_2/Ar)$ and NCP on the northern slope of the SCS was strongly affected by nutrient availability, especially nitrogen. The nitrogen limitation on NCP was found both in the autumn and summer. In June 2015, we observed strong biological responses to the supply of nitrogen induced by eddy-entrained shelf water intrusion. NCP in the region with the influence of shelf water was $23.8 \pm 10.7$ mmol C m$^{-2}$ d$^{-1}$ on average, with a maximum of 61.4 mmol C m$^{-2}$ d$^{-1}$. In addition, vertical mixing caused considerable underestimation of NCP in the transect influenced by a cold eddy. Removing the regions with the influence of shelf water intrusion and vertical mixing, the average NCP in other regions was $5.0 \pm 6.2$ mmol C m$^{-2}$ d$^{-1}$. This value agrees well with previously published NCP estimates for the study area. Our results also reveal the rapid response of ecosystem to physical processes. The summer shelf water intrusion may significantly promote NCP by potentially more than threefold in the study region. This is the first report that quantifies the contribution of shelf water intrusion to NCP on the northern slope of the SCS in the summer. Because of the sufficient illumination in the tropical SCS, light availability may not be a significant limitation on NCP in both seasons. The high-resolution NCP estimates derived from continuous measurement of $O_2/Ar$ presented in this paper are of significance for understanding the carbon cycle in the highly dynamic system of the SCS.

## Data Availability

All data presented in this manuscript are available on Weiyun.com (link: https://share.weiyun.com/ZtbQMNGl, password: p7rj36)

## Author contribution

Guiling Zhang and Yu Han designed and set up the underway measurement system. Wenjing Zheng attended both cruises (in June 2015 and October 2014) in the South China Sea, and was mainly responsible for operating the underway measurement system during the cruises. Sumei Liu provided the nutrients data of both cruises. Chuan Qin attended the cruise in June 2015 and prepared the manuscript with contributions from all co-authors.

## Acknowledgments

The authors wish to thank the crew of the *R/V* "Nanfeng" for the assistance with the collection of field samples and Professor Xiaoxia Sun for providing the [14]C-PP data. We would also like to thank the Ocean Biology Processing Group (OBPG) of NASA for generating the PAR data and the E.U. Copernicus Marine Environment Monitoring Service (CMEMS) for providing the satellite chlorophyll data. Professor Michael Bender and Bror Jonsson are acknowledged for constructive suggestions on the continuous $O_2/Ar$ measurement system and the calculation of $O_2/Ar$-based NCP. This study was funded by the National Science Foundation of China through Grant Nos. 41776122, by the Ministry of Science and Technology of China through Grant Nos. 2014CB441502, by the Fundamental Research Funds for the Central Universities (No. 201562010), and by the Taishan Scholars Programme of Shandong Province (No. 201511014) and the Aoshan Talents Programme of the Qingdao National Laboratory for Marine Science and Technology (No. 2015ASTP-OS08).

## Competing interests

The authors declare that they have no conflict of interest.

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

**Table Captions:**

**Table 1.** Basic information at all CTD stations in October 2014
**Table 2.** Basic information at all CTD stations in June 2015
**Table 3.** The start date and duration (Δday) of shelf water intrusion at the stations with surface
salinity lower than 33 in June 2015
**Table4.** Satellite-PAR data and NCP at the selected stations in October 2014

**Figure Captions:**

**Figure 1.** Cruise tracks of two cruises in the slope region of the Northern South China Sea in **(a)** October 2014, **(b)** June 2015. The sea level height anomaly (SLA) and geostrophic current during observations in June 2015 (Chen et al., 2016) are shown in **(c)**. The black dots/stars represent the locations of the CTD casts. Red numbers indicate transects, while black numbers indicate the serial number of CTD stations based on the cruise plan. The color scale in **(a)** and **(b)** represents bathymetry.

**Figure 2.** Surface distributions of **(a)** temperature, **(b)** salinity, **(c)** chlorophyll-a (Chl a), and **(d)** $\Delta(O_2/Ar)$ in October 2014

**Figure 3.** Surface distributions of **(a)** temperature, **(b)** salinity, **(c)** chlorophyll-a (Chl a), **(d)** dissolved oxygen (DO), **(e)** $pCO_2$, and **(f)** $\Delta(O_2/Ar)$ in June 2015

**Figure 4.** Surface distribution of NCP among the northern slope of SCS during the cruise in **(a)** October 2014 and **(b)** June 2015.

**Figure 5.** Zonal variations in **(a)** temperature, salinity, **(b)** $\Delta(O_2/Ar)$, **(c)** Chl a, NCP and surface concentration of ammonia ($NH_4^+$) along Transect 5 in October 2014. The plots of $\Delta(O_2/Ar)$ and NCP are 10-point Savitzky-Golay smoothed to give a better view of their distribution.

**Figure 6.** Meridional variations in **(a)** temperature, salinity, **(b)** $\Delta(O_2/Ar)$, $pCO_2$, **(c)** Chl a, NCP and surface concentration of DIN along Transect 4 in June 2015. The plots of $\Delta(O_2/Ar)$, $pCO_2$ and NCP are 10-point Savitzky-Golay smoothed.

**Figure 7.** Principal Component Analysis (PCA) among variables for **(a)** October 2014 and **(b)** June 2015 (Bartlett's test of sphericity: $p < 0.01$)

**Figure 8**. **(a)** T-S diagram of surface DIN concentration in June 2015. The stations influenced by shelf water were in the red circle. **(b)** Correlation analysis between surface DIN concentration and NCP at sampling stations. The stations (characterized with S < 33) influenced by shelf water presented surface DIN concentration $\geq 0.27$ $\mu$mol $L^{-1}$.

**Figure 9**. Correlation analysis between underway NCP and physical parameters (temperature and salinity) in October 2014 **(a, b)** and June 2015 **(c, d)**.

**Figure 10.** Daily satellite-chlorophyll images on the selected days in June 2015. Stars represent CTD locations. We roughly set satellite-chlorophyll $\geq 0.2$ $\mu$g $L^{-1}$ in this figure as the criterion of

shelf water. This figure was made based on the M_Map mapping package for MATLAB (Pawlowicz,

867    2020).

**Table 1.** Basic information at all CTD stations in October 2014

| Station | Date of observation[a] | MLD (m) | $Z_{eu}$[b] (m) | $k$[c] (m d$^{-1}$) | $\tau$[d] (d) |
|---|---|---|---|---|---|
| O-01 | 13 Oct 2014 | 58 | 82 | 4.7 | 12 |
| O-02 | 13 Oct 2014 | 64 | 74 | 5.2 | 12 |
| O-03 | 14 Oct 2014 | 56 | 84 | 6.2 | 9 |
| O-04 | 14 Oct 2014 | 54 | 76 | 6.3 | 9 |
| O-05 | 20 Oct 2014 | 27 | 70 | 7.9 | 3 |
| O-06 | 19 Oct 2014 | 55 | 62 | 8.4 | 7 |
| O-07 | 21 Oct 2014 | 40 | 60 | 7.3 | 5 |
| O-08 | 21 Oct 2014 | 49 | 72 | 7.4 | 7 |
| O-09 | 15 Oct 2014 | 79 | 96 | 6.2 | 13 |
| O-10 | 15 Oct 2014 | 68 | 81 | 6.1 | 11 |
| O-11 | 15 Oct 2014 | 64 | 81 | 5.4 | 12 |
| O-12 | 16 Oct 2014 | 66 | 74 | 5.2 | 13 |
| O-13 | 16 Oct 2014 | 48 | 52 | 6.3 | 8 |
| O-14 | 17 Oct 2014 | 54 | 62 | 6.9 | 8 |
| O-15 | 22 Oct 2014 | 49 | 68 | 7.0 | 7 |
| O-16 | 22 Oct 2014 | 50 | 73 | 7.3 | 7 |
| O-17 | 23 Oct 2014 | 52 | 75 | 7.9 | 7 |
| O-19 | 18 Oct 2014 | 31 | 64 | 9.4 | 3 |
| O-20 | 18 Oct 2014 | 35 | 61 | 8.7 | 4 |
| O-21 | 18 Oct 2014 | 81 | 86 | 6.9 | 12 |
| O-22 | 17 Oct 2014 | 76 | 102 | 6.0 | 13 |

[a] All dates are in the format of day month year. [b] Euphotic depth, defined based on subsurface chlorophyll maximum layer. [c] Gas transfer velocity of $O_2$. [d] Residence time of $O_2$ in the mixed layer, estimated as per MLD/k.


**Table 2.** Basic information at all CTD stations in June 2015

| Station | Date of observation | MLD (m) | $Z_{eu}$ (m) | k (m d$^{-1}$) | τ (d) |
|---|---|---|---|---|---|
| J-01 | 18 Jun 2015 | 26 | 63 | 2.2 | 12 |
| J-02 | 17 Jun 2015 | 19 | 80 | 1.9 | 10 |
| J-03 | 16 Jun 2015 | 20 | 74 | 1.9 | 11 |
| J-04 | 15 Jun 2015 | 22 | 74 | 1.9 | 11 |
| J-05 | 15 Jun 2015 | 11 | 78 | 1.2 | 9 |
| J-06 | 14 Jun 2015 | 24 | 76 | 2.1 | 11 |
| J-07 | 13 Jun 2015 | 21 | 81 | 2.3 | 9 |
| J-08 | 18 Jun 2015 | 14 | 56 | 1.7 | 8 |
| J-09 | 19 Jun 2015 | 17 | 59 | 1.6 | 10 |
| J-10 | 19 Jun 2015 | 8 | 46 | 1.4 | 6 |
| J-11 | 20 Jun 2015 | 8 | 40 | 2.8 | 3 |
| J-12 | 21 Jun 2015 | 16 | 45 | 3.0 | 5 |
| J-13 | 21 Jun 2015 | 19 | 45 | 2.3 | 8 |
| J-14 | 24 Jun 2015 | 28 | 55 | 4.0 | 7 |
| J-15 | 24 Jun 2015 | 17 | 42 | 5.3 | 3 |
| J-16 | 25 Jun 2015 | 10 | 19 | 5.7 | 2 |






**Table 3.**    The start date and duration ($\Delta$day) of shelf water intrusion at the stations with surface salinity lower than 33 in June 2015

| Station | Date of observation | Start date of shelf water intrusion | $\Delta$day[a] | $\tau$ (d) |
|---|---|---|---|---|
| J-09 | 19 Jun 2015 | 10 Jun 2015 | 9 | 10 |
| J-10 | 19 Jun 2015 | 13 Jun 2015 | 6 | 6 |
| J-11 | 20 Jun 2015 | 13 Jun 2015 | 7 | 3 |
| J-12 | 21 Jun 2015 | 13 Jun 2015 | 8 | 5 |
| J-13 | 21 Jun 2015 | 13 Jun 2015 | 8 | 8 |
| J-16 | 25 Jun 2015 | before 10 Jun 2015 | > 15 | 2 |

[a] The difference between the date of observation and the start date of shelf water intrusion at listed stations.

**Table 4.** Satellite-PAR data and NCP at the selected stations in October 2014

| Station | Date of observation | MLD (m) | $Z_{eu}$ (m) | Surface PAR[a] (mol m$^{-2}$ d$^{-1}$) | $K_d$ (m$^{-1}$) | ML PAR[b] (mol m$^{-2}$ d$^{-1}$) | NCP (mmol C m$^{-2}$ d$^{-1}$) |
|---------|--------------------|---------|--------------|-----------------------------------------|------------------|-------------------------------------|----------------------------------|
| O-01 | 13 Oct 2014 | 58 | 82 | 42.0 | $5.6 * 10^{-2}$ | 12.0 | 3.0 |
| O-02 | 13 Oct 2014 | 64 | 74 | 42.0 | $6.2 * 10^{-2}$ | 10.0 | 15.1 |
| O-03 | 14 Oct 2014 | 56 | 84 | 41.1 | $5.5 * 10^{-2}$ | 12.4 | 10.1 |
| O-08 | 21 Oct 2014 | 49 | 72 | 38.7 | $6.4 * 10^{-2}$ | 11.4 | 15.7 |
| O-10 | 15 Oct 2014 | 68 | 81 | 40.0 | $5.7 * 10^{-2}$ | 9.8 | 4.4 |
| O-13 | 16 Oct 2014 | 48 | 52 | 39.2 | $8.9 * 10^{-2}$ | 8.7 | 15.3 |
| O-15 | 22 Oct 2014 | 49 | 68 | 38.6 | $6.8 * 10^{-2}$ | 10.8 | 16.3 |
| O-20 | 18 Oct 2014 | 35 | 61 | 39.2 | $7.5 * 10^{-2}$ | 13.3 | 16.4 |
| O-22 | 17 Oct 2014 | 76 | 102 | 42.2 | $4.5 * 10^{-2}$ | 11.6 | 15.7 |

[a] Average surface PAR over the residence time of $O_2$ in the mixed layer. [b] Average PAR in the mixed layer.


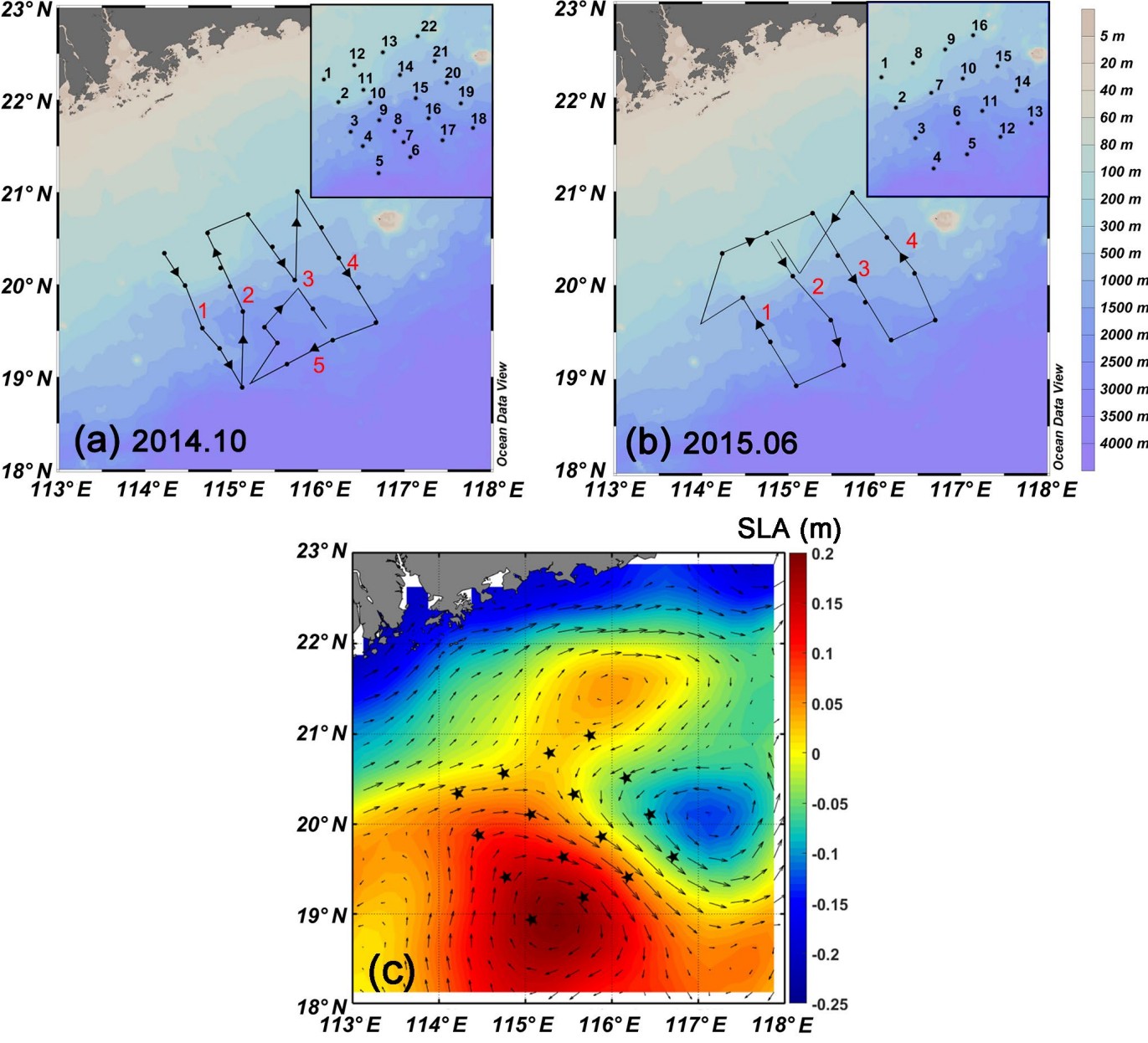

**Figure 1.** Cruise tracks of two cruises in the slope region of the Northern South China Sea in **(a)** October 2014, **(b)** June

2015. The sea level height anomaly (SLA) and geostrophic current during observations in June 2015 (Chen et al., 2016) are

shown in **(c)**. The black dots/stars represent the locations of the CTD casts. Red numbers indicate transects, while black

numbers indicate the serial number of CTD stations based on the cruise plan. The color scale in **(a)** and **(b)** represents

bathymetry.

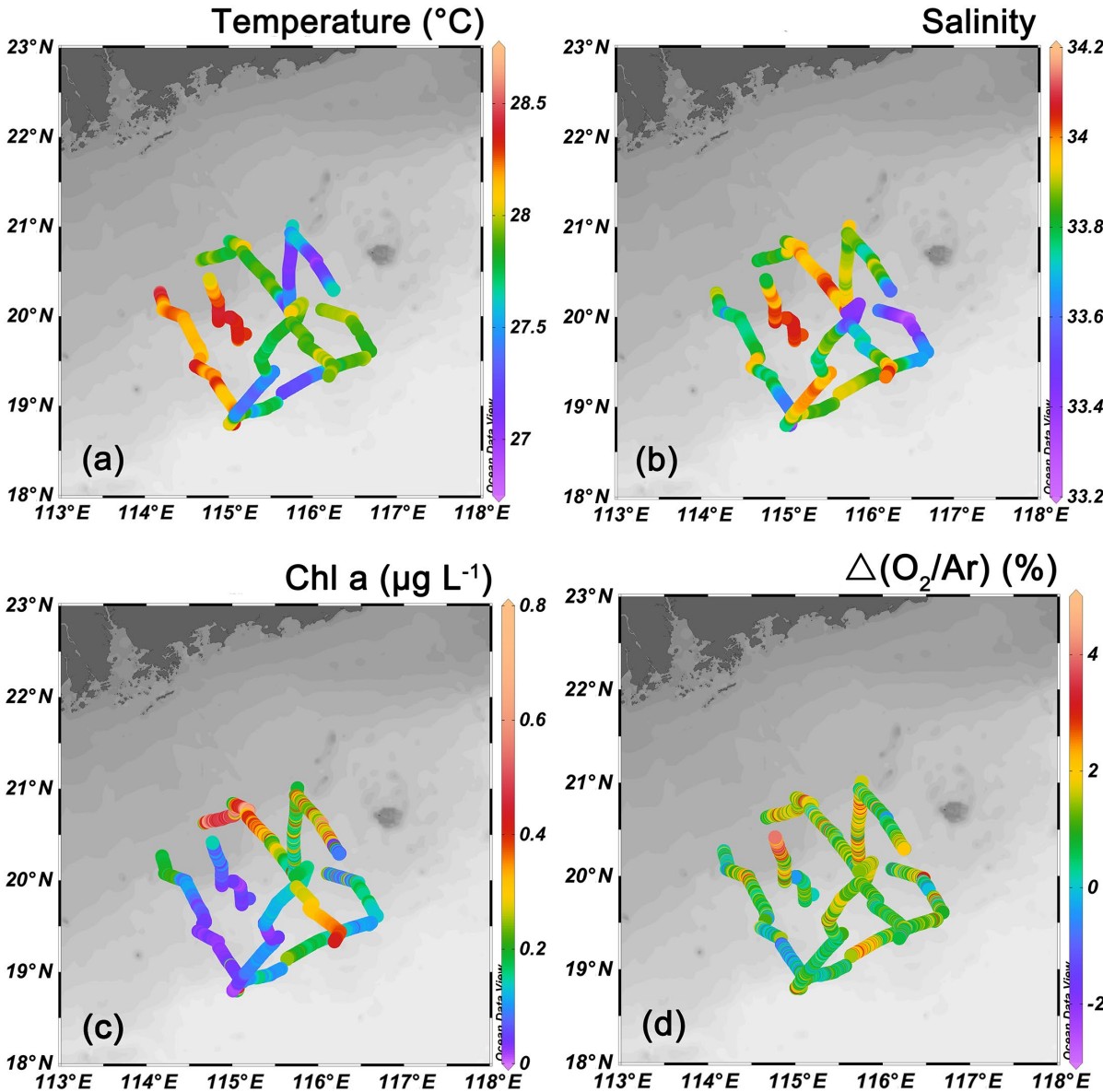


**Figure 2.** Surface distributions of **(a)** temperature, **(b)** salinity, **(c)** chlorophyll-a (Chl a), and **(d)** $\triangle(O_2/Ar)$ in October 2014


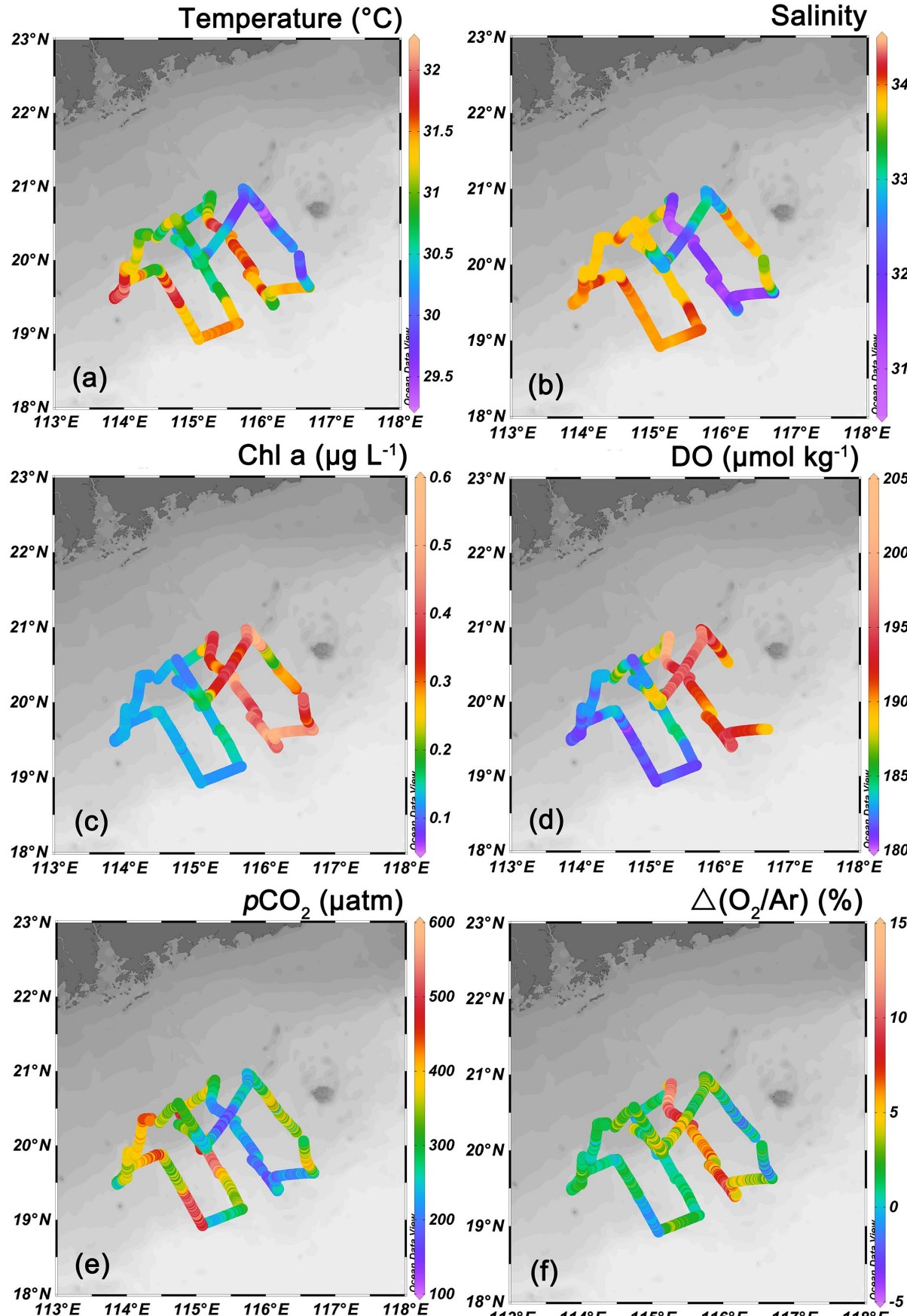

**Figure 3.** Surface distributions of **(a)** temperature, **(b)** salinity, **(c)** chlorophyll-a (Chl a), **(d)** dissolved oxygen (DO), **(e)** $p$CO$_2$, and **(f)** $\Delta$(O$_2$/Ar) in June 2015

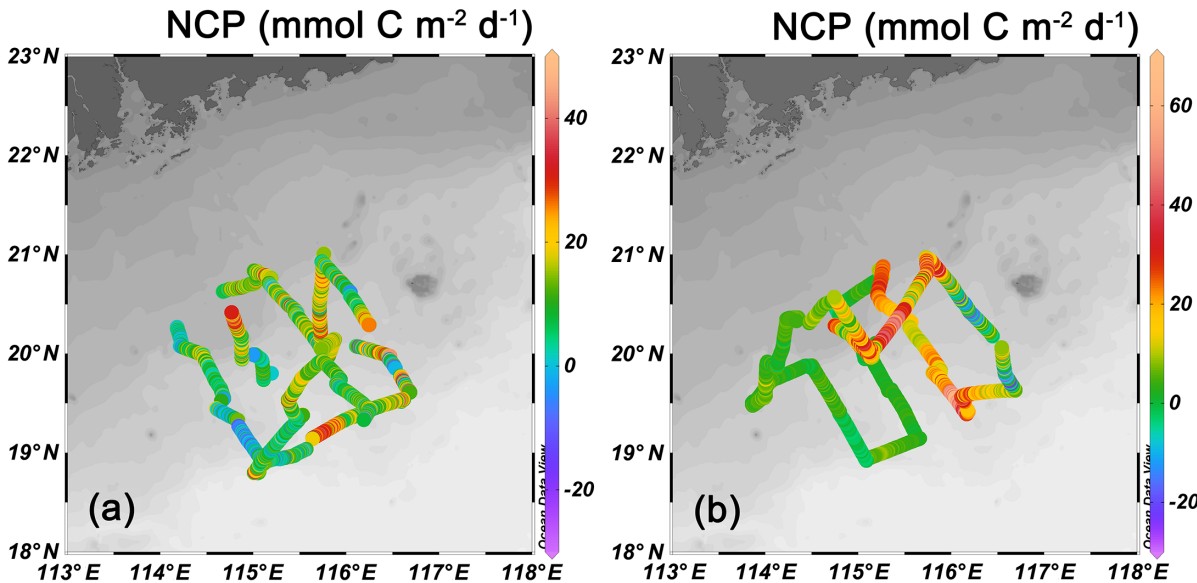


**Figure 4.** Surface distribution of NCP among the northern slope of SCS during the cruise in **(a)** October 2014 and **(b)** June

891 2015.



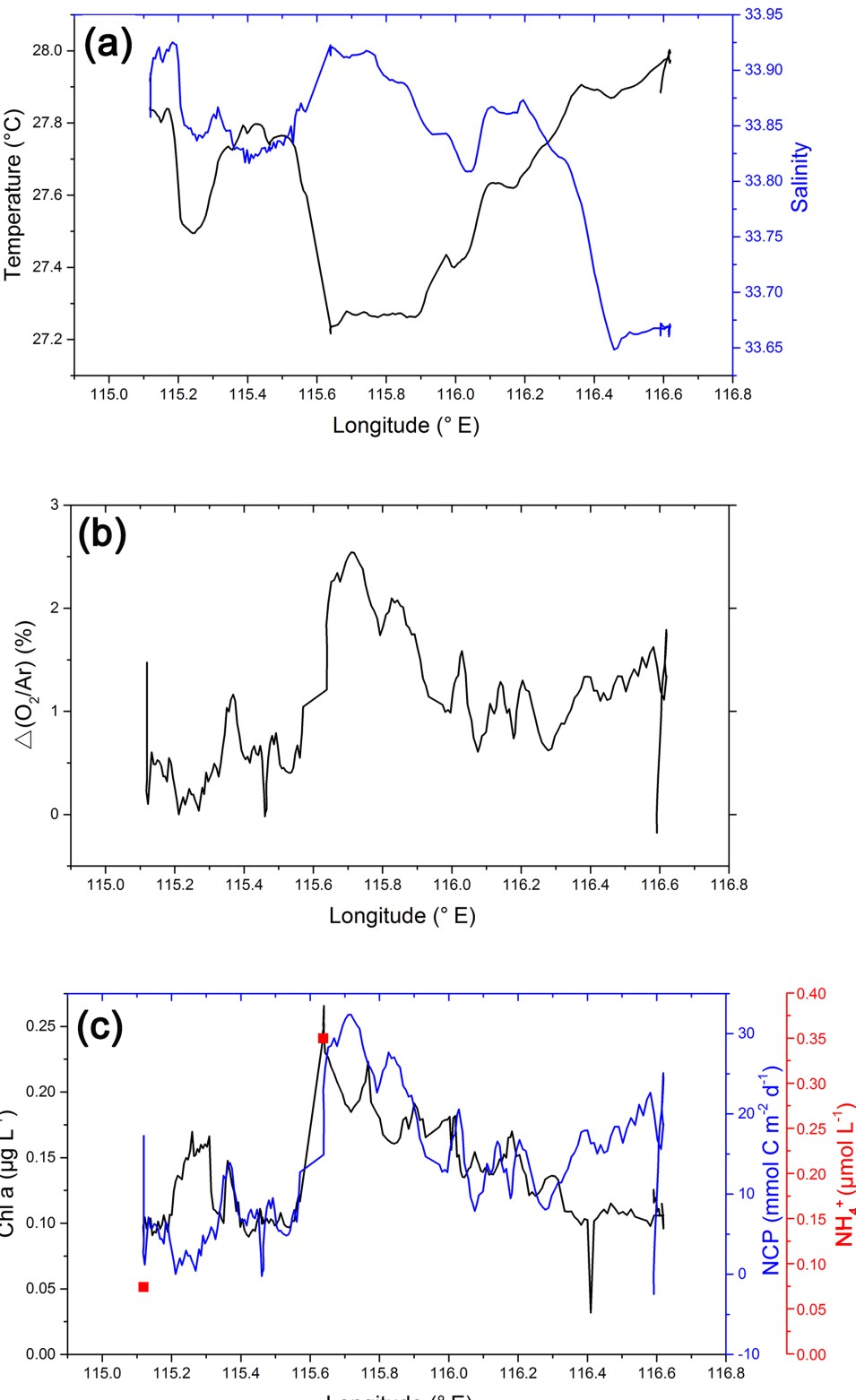


**Figure 5.** Zonal variations in **(a)** temperature, salinity, **(b)** $\Delta(O_2/Ar)$, **(c)** Chl a, NCP and surface concentration of ammonia

(NH$_4^+$) along Transect 5 in October 2014. The plots of $\Delta(O_2/Ar)$ and NCP are 10-point Savitzky-Golay smoothed to give a

better view of their distribution.

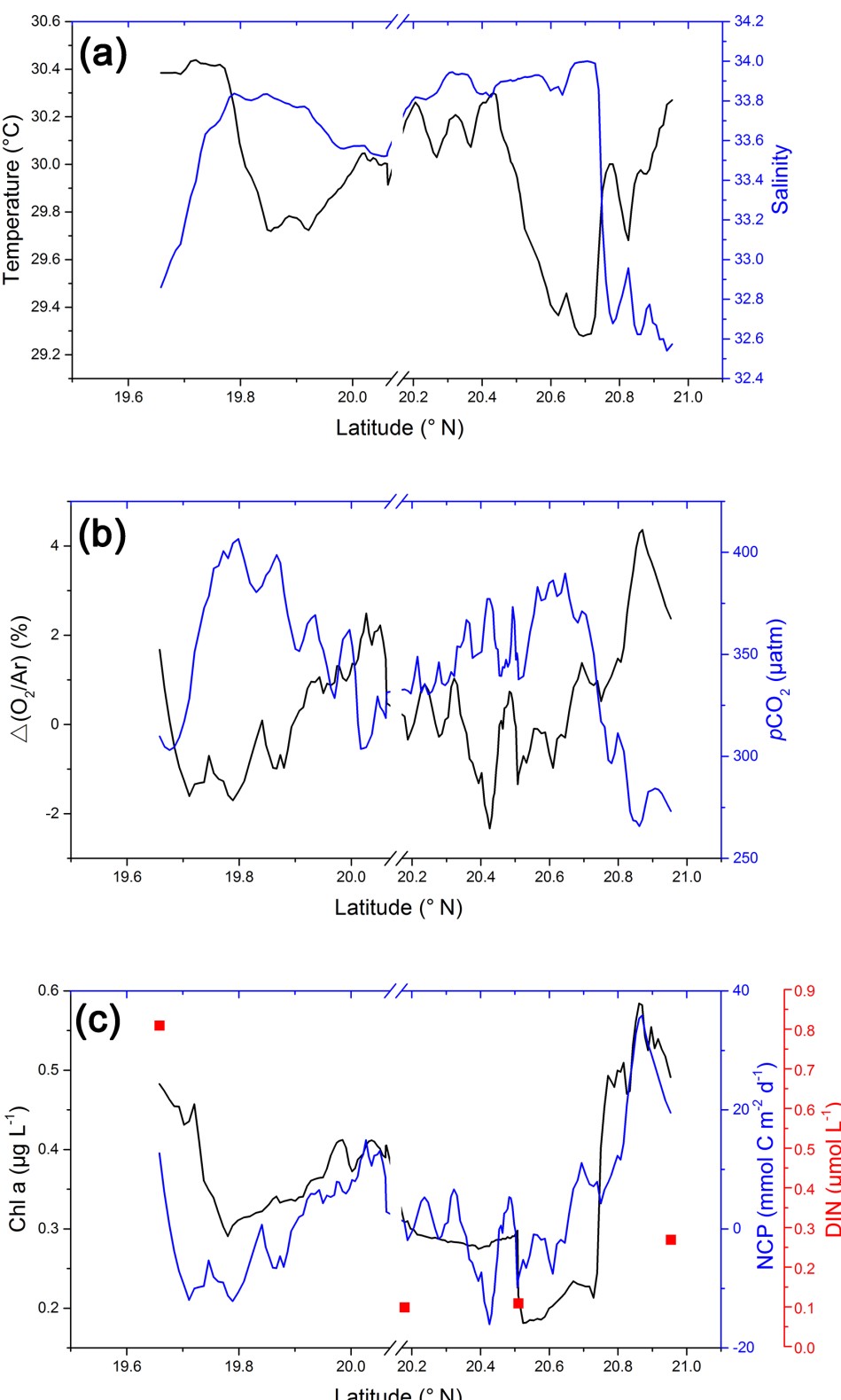

**Figure 6.** Meridional variations in **(a)** temperature, salinity, **(b)** $\Delta(O_2/Ar)$, $pCO_2$, **(c)** Chl a, NCP and surface concentration of DIN along Transect 4 in June 2015. The plots of $\Delta(O_2/Ar)$, $pCO_2$ and NCP are 10-point Savitzky-Golay smoothed.

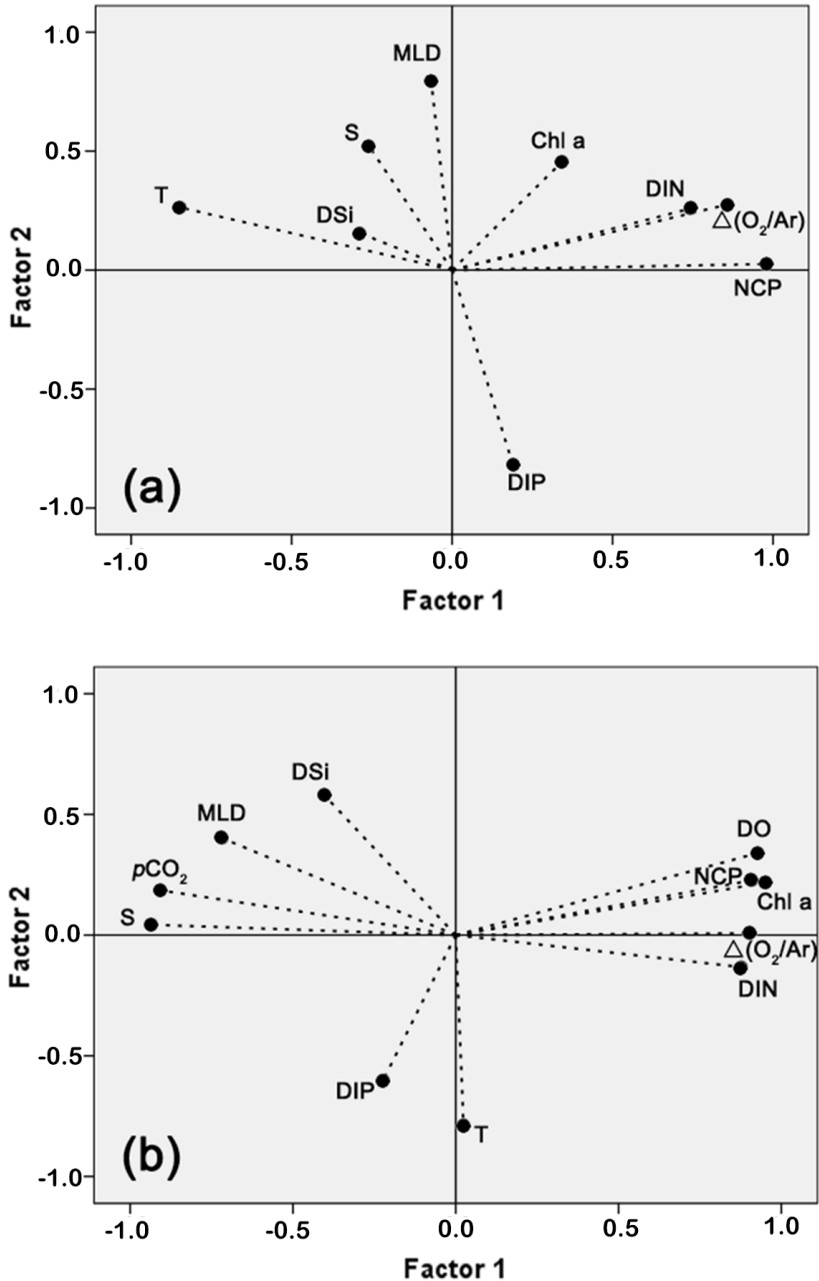

**Figure 7.** Principal Component Analysis (PCA) among variables for **(a)** October 2014 and **(b)** June 2015 (Bartlett's test of sphericity: $p < 0.01$)

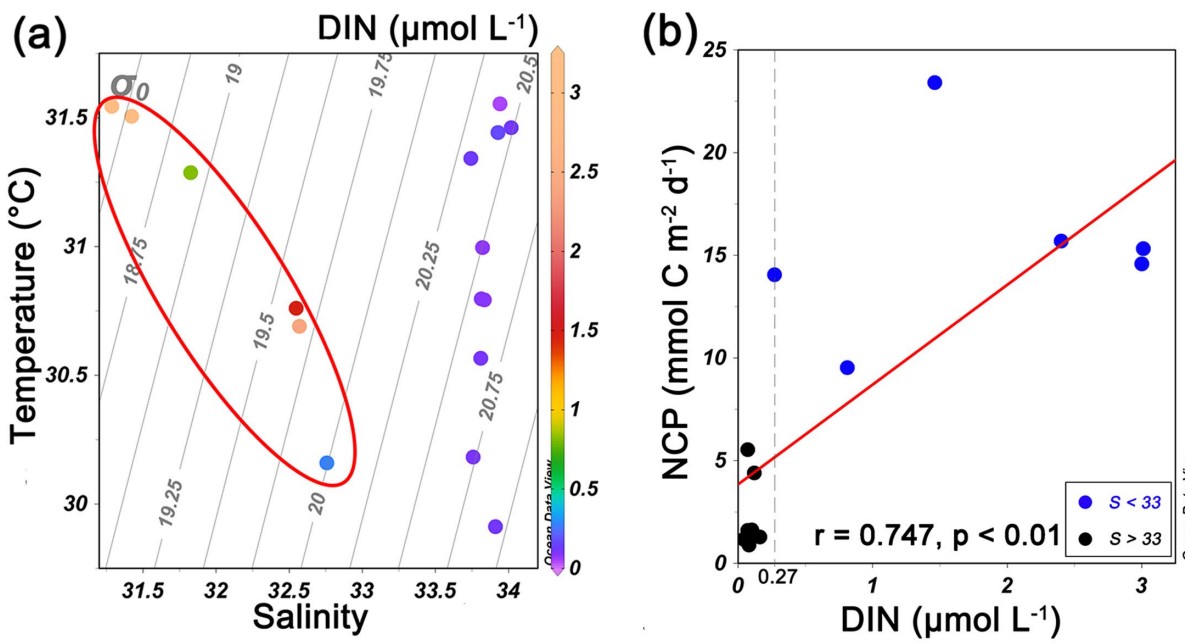


**Figure 8**. **(a)** T-S diagram of surface DIN concentration in June 2015. The stations influenced by shelf water were in the red

908        circle. **(b)** Correlation analysis between surface DIN concentration and NCP at sampling stations. The stations

909        (characterized with S < 33) influenced by shelf water presented surface DIN concentration $\geq 0.27$ µmol L$^{-1}$.



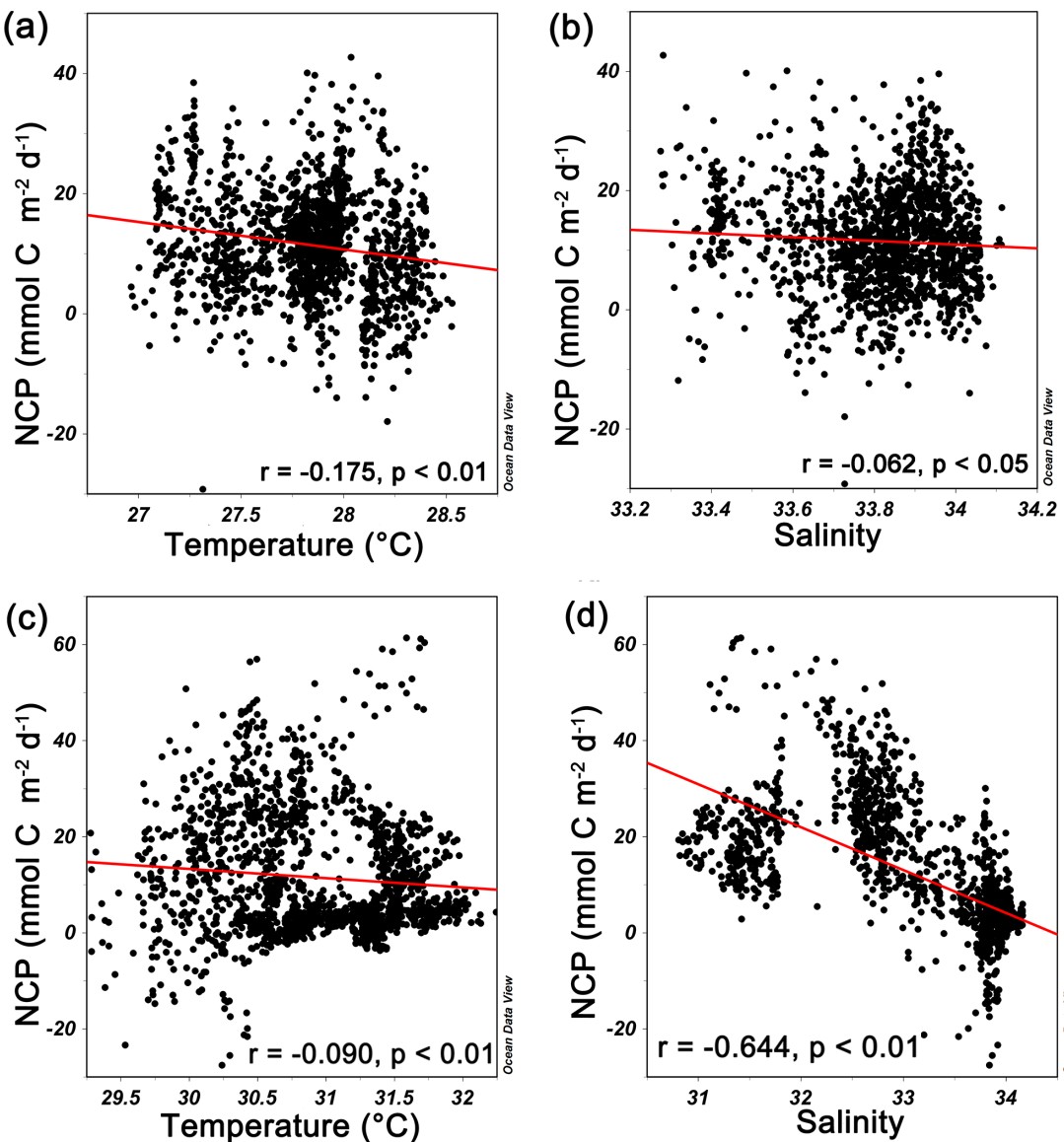


**Figure 9**. Correlation analysis between underway NCP and physical parameters (temperature and salinity) in October 2014

**(a, b)** and June 2015 **(c, d)**.


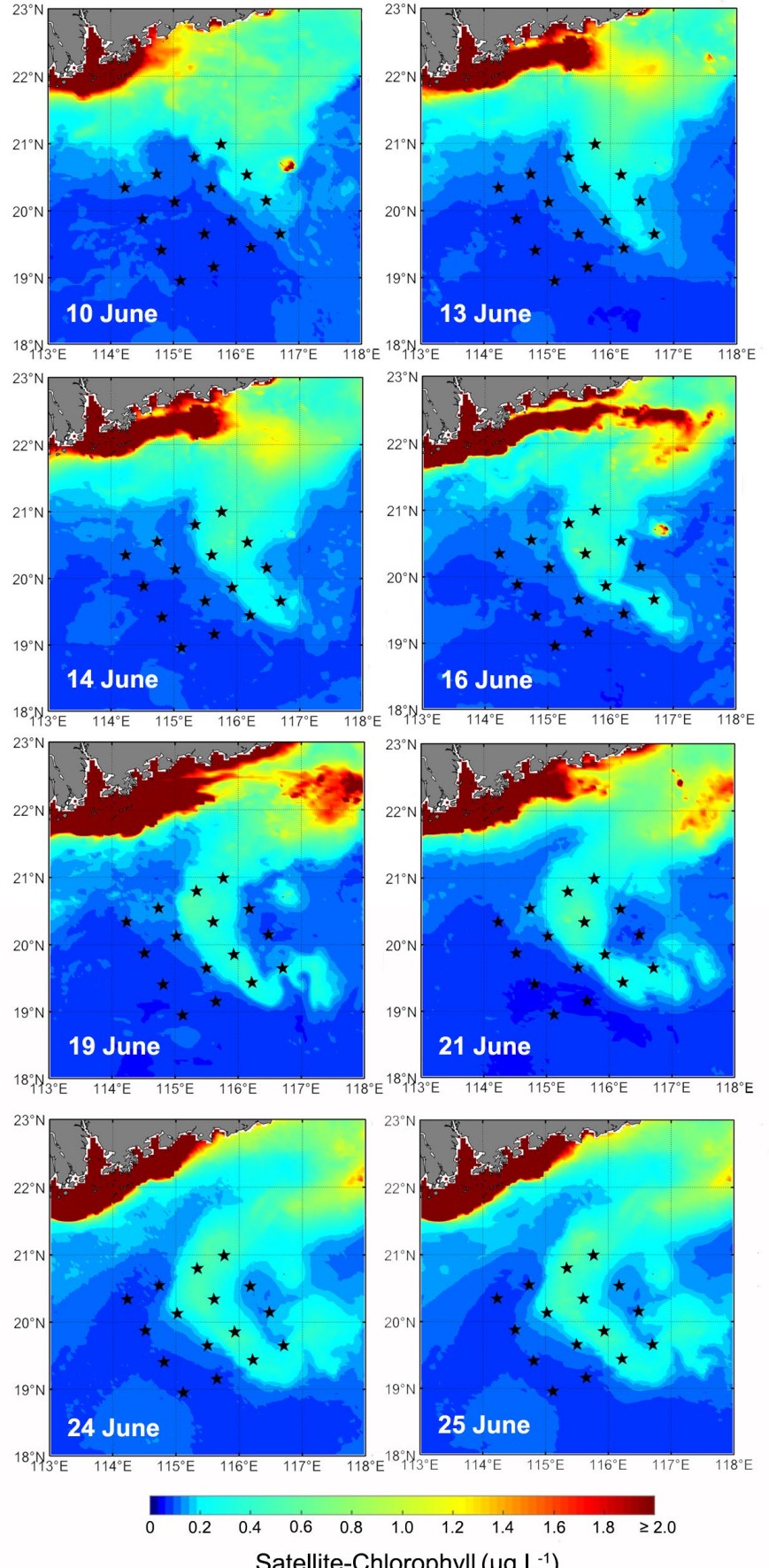


**Figure 10.** Daily satellite-chlorophyll images on the selected days in June 2015. Stars represent CTD locations. We roughly set satellite-chlorophyll ≥ 0.2 μg L$^{-1}$ in this figure as the criterion of shelf water. This figure was made based on the M_Map mapping package for MATLAB (Pawlowicz, 2020).