# Peer review of "High-resolution distributions of $\Delta(O_2/Ar)$ on the northern slope of the"

_Ocean Science, 2020_

## Referee Comment (RC1) · Anonymous Referee #1 · 16 Sep 2020

General comments:

This manuscript reports high-resolution measurements of net community production in the coastal South China Sea in summer and fall, as estimated using the dissolved O2/Ar technique. The authors compare measured NCP rates against physical parameters and nutrient concentrations to assess factors influencing the spatial distribution and magnitude of productivity.

These new NCP data are a useful contribution in that productivity measurements in this region to date have remained relatively sparse and low-resolution, but apart from presenting this new data the manuscript offers few broader or novel conclusions and

the wider scientific significance is limited.

The field, experimental, and statistical analyses appear to have been rigorously conducted largely according to current best practices. In particular the methods indicate a good experimental setup and attention to potential sources of error in field O2/Ar measurements.

In some cases, however, the conclusions regarding the influence of light and nutrients upon productivity patterns that the authors draw from their data and results are not fully appropriate or justified.

The presentation of data in the manuscript and figures is generally good, but leaves some room for improvement. The text could have benefited from better description of some data and variables, while Figure 1 is difficult to interpret, in turn impacting interpretation of other figures and results throughout the paper.

The major point this reviewer would like to stress is that the authors should pay careful attention to qualifying the caveats to some conclusions, while reconsidering other conclusions if they are not fully backed by the presented analyses.

1. The conclusion that NCP is subject to nitrogen limitation based on correspondences between elevated NCP rates and DIN concentrations needs further justification. Nutrient concentrations reflect the marine environment at the moment of sampling, while the O2/Ar method integrates over the residence time of biological oxygen in the surface ocean. The residence time is a particularly important factor for the authors to highlight in the manuscript, as it carries important implications for how far temporally removed the measured productivity signal is relative to the cruise measurements. At a minimum, additional discussion of the residence time of the oxygen signal on both cruises and potential associated considerations is necessary. Given the availability of satellite data as presented in He et al., 2016, discussion of the history of the measured water masses should be quite feasible. The reviewer also notes that the nitrogen limitation in this region is hardly a novel finding, as the authors themselves mention in the

introduction.

Mixed layer depth ranges and O2 surface layer residence time should be directly reported in the text of the results.

2. Similarly, assertions regarding the influence of light availability on NCP are questionable. Are MLDs around the time of these cruises really deep enough for light to influence mixed-layer productivity? In June of 2015 the data indicate that the maximum MLD value was just 30m. Water column PAR and Chl profiles are undoubtedly available from the CTD casts and should be presented and discussed in the context of such claims. A June 2015 Chl-a maxima of 0.6 ug/L certainly suggests that biomass-induced light attenuation in the mixed layer wouldn't be an issue, etc... Furthermore, residence time should once again be discussed, as light limitation is a factor influencing the time-integrated productivity signal over the relevant wind speed history at the measurement sites. Figure 10 also seems to suggest that the strength of MLD relationships with NCP are dependent on a relatively low number of data points.

3. The claim at lines 426-428 that "there was no significant productivity below the mixed layer that was missed by underway sampling" also seems unjustified and contradicts earlier results and text. Earlier for instance, the authors note that a subsurface oxygen maximum is a characteristic feature in the South China Sea, and significant subsurface productivity is observed in many oligotrophic regions globally such as the Sargasso Sea. Vertical ChI profiles would again be important evidence to present in support of the claim that underway measurements did not miss significant subsurface production.

4. In general, since Delta O2/Ar is directly used to calculate NCP, reporting and discussing relationships between Delta O2/Ar and other parameters provides little value. This reviewer would recommend removing discussions of Delta O2/Ar versus ChI, MLD, and so on from the manuscript.

5. The authors should also consider whether alternative hypotheses (nutrient colimitation, limitation by a non-measured variable such as Fe, etc...) could potentially

СЗ

present alternative explanations for observed patterns/relationships.

Specific comments:

Lines 13-14: it should be clarified at the very start of the abstract that the O2/Ar ratio refers to dissolved gases in surface seawater.

Line 32: Rather than oceanic CO2 uptake, which is strongly dominated by physical factors, it might be more appropriate to say that oceanic carbon sequestration is regulated by primary production and export.

Line 38: No longer recent... quite an established technique at this point.

Line 41: No longer clear that coastal O2/Ar-derived estimates of NCP are sparse.

Line 97: This is minor, but the original publication describing the MIMS technique (Tortell, L+O: Methods, 2005) should be cited here.

Line 158: Clarify units.

Lines 170-175: For this section, units for density should be in kg/m3 to arrive at the correct units for NCP, or the appropriate conversion factor

Line 181: remove "excellent"

Lines 213-220: Encourage authors to also display the ranges for SSS and Chl-a for the June 2015 cruise using the same format employed in the paragraph above for the October 2014 cruise, for consistency and clarity.

Lines 225-227: Elaborating on the assertion that high O2/Ar and low pCO2 signify a strong biological CO2 sink may be useful. The pCO2 values are indeed considerably low and I would be curious about the associated residence time of O2/Ar on Transect 3.

Line 236 and elsewhere: Here and throughout the paper, you are reporting averages as ##.## +/- ##.##. You should specify that these represent avg +/- std dev... etc...

Lines 299-301: Keep it clear in this sentence that upwelling does not necessarily produce an underestimation of NCP. It is more accurate to say that subsurface waters may have different (either more positive or more negative) O2/Ar signatures that could produce either an underestimation or an underestimation of NCP, as you explain a few sentences later in the context of subsurface O2 maxima in oligotrophic regions.

Lines 313-315: The cutoff for salinity to account for waters influenced by shelf water injection is justified, but the cutoff intended to exclude regions with upwelling seems somewhat arbitrary.

Line 330: Consideration of other explanations is needed in this section as described in the general comments.

Lines 412-418: As mentioned in the general comments, is it even meaningful to analyze the influence of light limitation on NCP when the euphotic zone is 2-7 times the depth of the mixed layer in this region? Much more discussion is needed to back this light limitation idea. Cassar et al 2011 makes the point for instance that MLD is not the only factor affecting light availability.

Lines 426-428: "This result implied that there was no significant productivity below the mixed layer that was missed by underway sampling." As mentioned in the general comments, you need to address the contradiction between this and subsurface O2 maxima and potential deep chlorophyll maxima.

Figure 1, lines 681 - 683: please describe what the point and star symbols represent. Are these the locations of CTD casts? The fact that the color scale in (a) and (b) represents bathymetry should also be stated as well. The cruise path in (a) is also difficult to interpret based on the arrows. This figure is critical for the interpretation of subsequent figures. Perhaps numbering of points on the cruise plan would convey the path of travel better.

Technical corrections:

Lines 310-318: Here and elsewhere throughout the manuscript, you use sentence structures that rely too heavily on parentheses, which can cause confusion for readers. These sentences in particular are very difficult to interpret.

General: There are some grammatical errors in the conclusion

---

## Referee Comment (RC2) · Anonymous Referee #2 · 16 Oct 2020

General comments

In their manuscript "High-resolution distributions of O2/Ar on the northern slope of the South China Sea and estimates of net community production", the authors report continuous net community production (NCP) estimates in the mixed layer of the northern South China Sea (SCS). The study makes a clear contribution to understanding of productivity in marginal seas like the SCS, where prior NCP estimates are limited. To a lesser extent, the study also advances a relatively novel method to estimating NCP through continuous observations of ∆O2/Ar. My major critique is that the authors do not connect back to these original objectives in their paper. What new information have

they gathered about the SCS as a result of this continuous method of measuring NCP, and how does this relate to past measurements of NCP in this region? Which methodological and/or environmental factors cause their estimates to compare or differ from past estimates? It is clear why their study is significant, but explicitly tying the discussion of results to these objectives will strengthen the scientific contributions of this paper.

Specific comments

In the title: It is more accurate to write $\Delta O_2/Ar$ rather than $O_2/Ar$?

Line 35: Clarify what "indicator" means in this context, and in which conditions this assumption holds true (e.g., NCP may be partitioned into DOC production, particle export, zooplankton grazing, etc.).

Line 55: The water classifications are a bit confusing here. Perhaps it would be clearer to say that SCS water is a mix between two end-members: freshwater runoff from rivers and North Pacific offshore water.

Lines 70-84: It is unclear what the aim of listing these numbers is? Do the authors wish to convey that NCP is variable across SCS studies? It would be useful to reference these numbers again in the discussion for comparison. In any case, when reporting NCP and export, use both O2 and C units so that the numbers are comparable. The authors can perhaps apply the photosynthetic quotient used in the method to do this conversion to keep units consistent (line 174).

Line 79: Should the units here be $s^{-1}$ rather than $a^{-1}$?

Lines 85-86: Describe the potential inaccuracies of each discrete method so that it is clearer how this study benefits scientific understanding of the SCS.

Section 2.2: Explain how the 5-minute NCP values are scaled up to daily estimates.

Lines 179-183: As written, these two sentences imply that the authors do not know

whether their NCP estimates represent daily or monthly signals. If they represent the latter, would this not defeat the purpose of the study, which is to resolve "highly dynamic environmental fluctuations of coastal systems" (line 87) in shorter than monthly time scales?

Lines 412-416: Clarify how the DIN and NCP criteria were chosen for each cruise. Fig. 10b is not very compelling as the lowest MLD - highest volumetric NCP data point seems to drive the negative correlation. Thus, the authors should consider removing their analysis of June 2015 data from Fig. 10b, and just discuss the analysis in the text in relation to the much stronger relationship between MLD and volumetric NCP during October 2014. Another related analysis that may be interesting is comparing NCP values at stations where MLD is deeper than the euphotic zone depth, to NCP at stations where the MLD is shallower than the euphotic zone depth.

Technical comments

Figure 1: Explain what the dots/markers in the panels represent. Are they the locations of the CTD casts? If not, it is worth adding the locations of the CTD casts to this figure so that readers may better understand the interpolation of MLD between casts for underway data.

Figure 3: Why were there more variables in the June cruise? This is not clear in the methods.

Figure 5: Write in the salinity units. It is worth clarifying somewhere in the figure text, as well as the main text referencing Fig. 5, that the temperature fluctuations shown here are too small to reflect upwelling.

Figure 6: Write in the salinity units.

Figure 8: Write in the salinity units.

Figure 9: Write in the salinity units.

Figure S1: This actually is referenced after Fig. S2 so consider switching the figure order. Why is [NO3-] omitted here?

Figures S3-S7: These are not referenced in the text, but they should be if they are to be published. Otherwise, it is not clear what the significance of showing these data are, as they could just go on an online repository which gets referenced in the text.

---

## Author Response (AR1)

**Reply to Referee #1**

General comments:

This manuscript reports high-resolution measurements of net community production in the coastal South China Sea in summer and fall, as estimated using the dissolved $O_2/Ar$ technique. The authors compare measured NCP rates against physical parameters and nutrient concentrations to assess factors influencing the spatial distribution and magnitude of productivity. These new NCP data are a useful contribution in that productivity measurements in this region to date have remained relatively sparse and low-resolution, but apart from presenting this new data the manuscript offers few broader or novel conclusions and the wider scientific significance is limited.

The field, experimental, and statistical analyses appear to have been rigorously conducted largely according to current best practices. In particular the methods indicate a good experimental setup and attention to potential sources of error in field $O_2/Ar$ measurements.

In some cases, however, the conclusions regarding the influence of light and nutrients upon productivity patterns that the authors draw from their data and results are not fully appropriate or justified.

The presentation of data in the manuscript and figures is generally good, but leaves some room for improvement. The text could have benefited from better description of some data and variables, while Figure 1 is difficult to interpret, in turn impacting interpretation of other figures and results throughout the paper.

The major point this reviewer would like to stress is that the authors should pay careful attention to qualifying the caveats to some conclusions, while reconsidering other conclusions if they are not fully backed by the presented analyses.

**Response:** Thank you very much for your constructive comments. We have considered all suggestions and incorporated them into the revised version. In the following we answer to your comments point by point and indicate how the manuscript is going to be revised.

1. The conclusion that NCP is subject to nitrogen limitation based on correspondences between elevated NCP rates and DIN concentrations needs further justification. Nutrient concentrations reflect the marine environment at the moment of sampling, while the $O_2/Ar$ method integrates over the residence time of biological oxygen in the surface ocean. The residence time is a particularly important factor for the authors to highlight in the manuscript, as it carries important implications for how far temporally removed the measured productivity signal is relative to the cruise measurements. At a minimum, additional discussion of the residence time of the oxygen signal on both cruises and potential associated considerations is necessary. Given the availability of satellite data as presented in He et al., 2016, discussion of the history of the measured water masses should be quite feasible. The reviewer also notes that the nitrogen limitation in this region is hardly a novel finding, as the authors themselves mention in the introduction. Mixed layer depth ranges and $O_2$ surface layer residence time should be directly reported in the text of the results.

**Response:** Thanks for your suggestion. We have reported the residence time of $O_2$ in the mixed layer in both cruises together with other parameters such as mixed layer depth and euphotic depth by adding two tables (Table 1 and 2), and incorporated them into the results and discussion. We have also applied the satellite-Chl a data (https://resources.marine.copernicus.eu) to track the history of shelf water intrusion and combined it with the residence time of $O_2$ to strengthen the reliability of the relationship between NCP and nitrogen we observed. Table 3 and Figure S3 were added in the new version accordingly. Δday in Table 3 is the difference between the date of observation and the start date of shelf water intrusion at stations with surface salinity lower than 33, representing the duration of the shelf water intrusion at each station before our observation. The mixed layer $O_2$ residence time at most stations listed in Table 3 is shorter than or equivalent to Δday. This result suggests that our estimate has appropriately integrated the NCP during the period of shelf water intrusion, which can effectively reflect the influence of shelf water on productive state on the northern slope of SCS in the summer.

The nitrogen limitation in SCS has been reported before, but so far no research has focused on quantifying the NCP enhancement caused by shelf water intrusion in this region. Our research is also the first report that quantifies the contribution of shelf water intrusion to NCP on the northern slope of the SCS in the summer. We have highlighted this in the introduction and conclusion section.

**Table 1.** Basic information at all CTD stations in October 2014

| Station | Date of observation[a] | MLD (m) | $Z_{eu}$[b] (m) | k[c] (m d$^{-1}$) | $\tau$[d] (d) |
|---|---|---|---|---|---|
| O-01 | 2014/10/13 | 58 | 82 | 4.7 | 12 |
| O-02 | 2014/10/13 | 64 | 74 | 5.2 | 12 |
| O-03 | 2014/10/14 | 56 | 84 | 6.2 | 9 |
| O-04 | 2014/10/14 | 54 | 76 | 6.3 | 9 |
| O-05 | 2014/10/20 | 27 | 70 | 7.9 | 3 |
| O-06 | 2014/10/19 | 55 | 62 | 8.4 | 7 |
| O-07 | 2014/10/21 | 40 | 60 | 7.3 | 5 |
| O-08 | 2014/10/21 | 49 | 72 | 7.4 | 7 |
| O-09 | 2014/10/15 | 79 | 96 | 6.2 | 13 |
| O-10 | 2014/10/15 | 68 | 81 | 6.1 | 11 |
| O-11 | 2014/10/15 | 64 | 81 | 5.4 | 12 |
| O-12 | 2014/10/16 | 66 | 74 | 5.2 | 13 |
| O-13 | 2014/10/16 | 48 | 52 | 6.3 | 8 |
| O-14 | 2014/10/17 | 54 | 62 | 6.9 | 8 |
| O-15 | 2014/10/22 | 49 | 68 | 7.0 | 7 |
| O-16 | 2014/10/22 | 50 | 73 | 7.3 | 7 |
| O-17 | 2014/10/23 | 52 | 75 | 7.9 | 7 |
| O-19 | 2014/10/18 | 31 | 64 | 9.4 | 3 |
| O-20 | 2014/10/18 | 35 | 61 | 8.7 | 4 |
| O-21 | 2014/10/18 | 81 | 86 | 6.9 | 12 |
| O-22 | 2014/10/17 | 76 | 102 | 6.0 | 13 |

[a] All dates are in the format of year/month/day. [b] Euphotic depth, defined based on subsurface chlorophyll maximum layer. [c] Gas transfer velocity of $O_2$. [d] Residence time of $O_2$ in the mixed layer, estimated as per MLD/k.

**Table 2.** Basic information at all CTD stations in June 2015

| Station | Date of observation | MLD (m) | $Z_{eu}$ (m) | k (m d$^{-1}$) | $\tau$ (d) |
|---------|---------------------|---------|--------------|----------------|-----------|
| J-01 | 2015/6/18 | 26 | 63 | 2.2 | 12 |
| J-02 | 2015/6/17 | 19 | 80 | 1.9 | 10 |
| J-03 | 2015/6/16 | 20 | 74 | 1.9 | 11 |
| J-04 | 2015/6/15 | 22 | 74 | 1.9 | 11 |
| J-05 | 2015/6/15 | 11 | 78 | 1.2 | 9 |
| J-06 | 2015/6/14 | 24 | 76 | 2.1 | 11 |
| J-07 | 2015/6/13 | 21 | 81 | 2.3 | 9 |
| J-08 | 2015/6/18 | 14 | 56 | 1.7 | 8 |
| J-09 | 2015/6/19 | 17 | 59 | 1.6 | 10 |
| J-10 | 2015/6/19 | 8 | 46 | 1.4 | 6 |
| J-11 | 2015/6/20 | 8 | 40 | 2.8 | 3 |
| J-12 | 2015/6/21 | 16 | 45 | 3.0 | 5 |
| J-13 | 2015/6/21 | 19 | 45 | 2.3 | 8 |
| J-14 | 2015/6/24 | 28 | 55 | 4.0 | 7 |
| J-15 | 2015/6/24 | 17 | 42 | 5.3 | 3 |
| J-16 | 2015/6/25 | 10 | 19 | 5.7 | 2 |

**Table 3.** The start date and duration ($\Delta$day) of shelf water intrusion at stations with surface salinity lower than 33 in June 2015

| Station | Date of observation | Start date of shelf water intrusion | $\Delta$day[a] | $\tau$ (d) |
|---------|---------------------|--------------------------------------|----------------|-----------|
| J-09 | 2015/6/19 | 2015/6/10 | 9 | 10 |
| J-10 | 2015/6/19 | 2015/6/13 | 6 | 6 |
| J-11 | 2015/6/20 | 2015/6/13 | 7 | 3 |
| J-12 | 2015/6/21 | 2015/6/13 | 8 | 5 |
| J-13 | 2015/6/21 | 2015/6/13 | 8 | 8 |
| J-16 | 2015/6/25 | before 2015/6/10 | > 15 | 2 |

[a] The difference between the date of observation and the start date of shelf water intrusion at listed stations.

[Figure]

**Figure S3.** Daily satellite-chlorophyll on selected date from June 10 to 25, 2015. Stars represent CTD locations. We roughly set light blue (represents ~ 0.2 µg L$^{-1}$) in this figure as the criterion of shelf water

2. Similarly, assertions regarding the influence of light availability on NCP are questionable. Are MLDs around the time of these cruises really deep enough for light to influence mixed-layer productivity? In June of 2015 the data indicate that the maximum MLD value was just 30m. Water column PAR and Chl profiles are undoubtedly available from the CTD casts and should be presented and discussed in the context of such claims. A June 2015 Chl-a maxima of 0.6 ug/L certainly suggests that biomass-induced light attenuation in the mixed layer wouldn't be an issue, etc: : : Furthermore, residence time should once again be discussed, as light limitation is a factor influencing the time integrated productivity signal over the relevant wind speed history at the measurement sites. Figure 10 also seems to suggest that the strength of MLD relationships with NCP are dependent on a relatively low number of data points.

**Response:** Thanks for your suggestion. Because of your comment and the comments of reviewer 2, we have noticed the limitation of our analysis of MLD and $NCP_{vol}$. The negative correlation between $NCP_{vol}$ and MLD we obtained may partly result from that $NCP_{vol}$ is calculated by NCP/MLD. Thus we calculate an average surface PAR by integrating the daily satellite-PAR data obtained from NASA's ocean color website (https://oceancolor.gsfc.nasa.gov/l3) over the residence time of $O_2$ at each selected station in October 2014. 8-day PAR data were used to estimate the missing daily data. Then we use light attenuation coefficient ($K_d$) to calculate an average PAR in the mixed layer to make a correlation analysis with NCP. This new analysis gives a result that light availability is not a limitation on NCP in the SCS (Table 4), much more convincing than the former analysis just based on MLD.

The calculation of $K_d$ basically based on Lambert-Beer law (Kirk 1994; Jerlov 1976):

$$K_d = -\frac{1}{z} \ln \frac{E_d(z)}{E_d(0)} = \frac{4.605}{Z_{eu}}$$

Where $K_d$ ($m^{-1}$) is the light attenuation coefficient in the euphotic layer; $E_d(0)$ is the PAR at the surface, integrating an average over the residence time of $O_2$ before our observation, in the unit of mol $m^{-2}$ $d^{-1}$; z represents a depth (m) and $E_d(z)$ is the PAR at this depth; $Z_{eu}$ is the euphotic depth (m).

**References:** Kirk, J. T.: Light and photosynthesis in aquatic ecosystems, Cambridge university press, UK, 1994.

Jerlov, N. G.: Marine optics, Elsevier, Netherlands, 1976.

**Table 4.** Satellite-PAR data and NCP at selected stations in October 2014

| Station | Date of observation | MLD (m) | $Z_{eu}$ (m) | Surface PAR[a] (mol $m^{-2}$ $d^{-1}$) | $K_d$ ($m^{-1}$) | ML PAR[b] (mol $m^{-2}$ $d^{-1}$) | NCP (mmol C $m^{-2}$ $d^{-1}$) |
|---|---|---|---|---|---|---|---|
| O-01 | 2014/10/13 | 58 | 82 | 42.0 | $5.6 * 10^{-2}$ | 12.0 | 3.0 |
| O-02 | 2014/10/13 | 64 | 74 | 42.0 | $6.2 * 10^{-2}$ | 10.0 | 15.1 |
| O-03 | 2014/10/14 | 56 | 84 | 41.1 | $5.5 * 10^{-2}$ | 12.4 | 10.1 |
| O-08 | 2014/10/21 | 49 | 72 | 38.7 | $6.4 * 10^{-2}$ | 11.4 | 15.7 |
| O-10 | 2014/10/15 | 68 | 81 | 40.0 | $5.7 * 10^{-2}$ | 9.8 | 4.4 |
| O-13 | 2014/10/16 | 48 | 52 | 39.2 | $8.9 * 10^{-2}$ | 8.7 | 15.3 |
| O-15 | 2014/10/22 | 49 | 68 | 38.6 | $6.8 * 10^{-2}$ | 10.8 | 16.3 |
| O-20 | 2014/10/18 | 35 | 61 | 39.2 | $7.5 * 10^{-2}$ | 13.3 | 16.4 |
| O-22 | 2014/10/17 | 76 | 102 | 42.2 | $4.5 * 10^{-2}$ | 11.6 | 15.7 |

[a] Average surface PAR over the residence time of $O_2$ in the mixed layer. [b] Average PAR in the mixed layer.

3. The claim at lines 426-428 that "there was no significant productivity below the mixed layer that was missed by underway sampling" also seems unjustified and contradicts earlier results and text. Earlier for instance, the authors note that a subsurface oxygen maximum is a characteristic feature in the South China Sea, and significant subsurface productivity is observed in many oligotrophic regions globally such as the Sargasso Sea. Vertical Chl profiles would again be important evidence to present in support of the claim that underway measurements did not miss significant subsurface production.

**Response:** Sorry for this arbitrary conclusion. We have deleted it. Subsurface chlorophyll maximum layer (SCML) existed at all stations in both cruises, while the obvious subsurface $O_2$ maximum just existed in Transect 1 and 2 in June 2015. We have reported these results in section 3.2. Both SCML and subsurface $O_2$ maximum were below the MLD, indicating that they might not influence the NCP in the mixed layer a lot.

4. In general, since Delta $O_2$/Ar is directly used to calculate NCP, reporting and discussing relationships between Delta $O_2$/Ar and other parameters provides little value. This reviewer would recommend removing discussions of Delta $O_2$/Ar versus Chl, MLD, and so on from the manuscript.

**Response:** Thanks for your suggestion. We have done that and also removed related figures from Figure 8 and 9.

5. The authors should also consider whether alternative hypotheses (nutrient co-limitation, limitation by a non-measured variable such as Fe, etc: : :) could potentially present alternative explanations for observed patterns/relationships.

**Response:** Thanks for your suggestion. We had tried to analyze the relationships between NCP and Fe, N:P ratio, but we didn't find any valuable result to report. Zhang et al. (2019) reported that Fe may not be a limitation to phytoplankton growth on the northern slope of SCS. In addition, we didn't find significant correlation between Fe (data provided by Ruifeng Zhang) and NCP in the mixed layer in Oct. 2014, thus we didn't report this result. N is acknowledged as the major limitation of primary production in the SCS, and P deficiency usually occurs when excessive river runoff results in a high N:P ratio nutritive state (Lee Chen and Chen 2006). N:P ratio is an important basis for judging whether the main influencing factor on NCP is N or P in the SCS. But most of our nutrients data in the mixed layer are too low to support a convincing analysis of N:P ratio. We have added some background information of these alternative hypotheses in the introduction section.

**References:** Zhang, R., Zhu, X., Yang, C., Ye, L., Zhang, G., Ren, J., Wu, Y., Liu, S., Zhang, J. and Zhou, M.: Distribution of dissolved iron in the Pearl River (Zhujiang) Estuary and the northern continental slope of the South China Sea, Deep. Res. Part II Top. Stud. Oceanogr., 167, 14–24, doi:10.1016/j.dsr2.2018.12.006, 2019.

Lee Chen, Y. and Chen, H.: Seasonal dynamics of primary and new production in the northern South China Sea: The significance of river discharge and nutrient advection, Deep Sea Res. Part I Oceanogr. Res. Pap., 53(6), 971–986, doi:10.1016/j.dsr.2006.02.005, 2006.)

**Specific comments:**

Lines 13-14: it should be clarified at the very start of the abstract that the $O_2$/Ar ratio refers to dissolved gases in surface seawater.

**Response:** Following your suggestion, we started the abstract with *"Dissolved oxygen-to-argon ratios ($O_2$/Ar) in the oceanic mixed layer has been widely used to estimate net community production (NCP), which is the difference between gross primary production and community respiration and is a proxy of carbon export from the surface ocean."*.

Line 32: Rather than oceanic $CO_2$ uptake, which is strongly dominated by physical factors, it might be more appropriate to say that oceanic carbon sequestration is regulated by primary production and export.
**Response:** Thanks for your suggestion. We have changed the oceanic $CO_2$ uptake to oceanic carbon sequestration.

Line 38: No longer recent: : : quite an established technique at this point.
**Response:** We agree with the reviewer that using the $O_2/Ar$ ratio to estimate NCP is an established technique, hence the word 'Recently' was deleted.

Line 41: No longer clear that coastal $O_2/Ar$-derived estimates of NCP are sparse.
**Response:** Thanks for your suggestion. We have revised this paragraph and it now reads as *"During recent years, several high-resolution measurements of $O_2/Ar$ and NCP in coastal waters have been reported (Tortell et al., 2012; Tortell et al., 2014; Eveleth et al., 2017; Izett et al., 2018). Despite the coastal waters such as shelves and estuaries only account for 7 % of the global ocean surface area, they are known to contribute to 15–30 % of the total oceanic primary production (Bi et al., 2013; Cai et al., 2011) and play an important role in marine carbon cycle and production. However, these regions still suffer from low resolution measurements and are poorly represented in global NCP data sets."*

Line 97: This is minor, but the original publication describing the MIMS technique (Tortell, L&O: Methods, 2005) should be cited here.
**Response:** Thanks for your suggestion. We have cited this paper here.

Line 158: Clarify units.
**Response:** The unit here is $\mu g\ L^{-1}$.

Lines 170-175: For this section, units for density should be in kg/m^3 to arrive at the correct units for NCP, or the appropriate conversion factor
**Response:** We have revised it as suggested.

Line 181: remove "excellent"
**Response:** We have deleted it.

Lines 213-220: Encourage authors to also display the ranges for SSS and Chl-a for the June 2015 cruise using the same format employed in the paragraph above for the October 2014 cruise, for consistency and clarity.
**Response:** Thanks for your suggestion. We have done that as suggested.

Lines 225-227: Elaborating on the assertion that high $O_2/Ar$ and low $pCO_2$ signify a strong biological $CO_2$ sink may be useful. The $pCO_2$ values are indeed considerably low and I would be curious about the associated residence time of $O_2/Ar$ on Transect 3.
**Response:** Thanks for your suggestion. We have applied an atmospheric $pCO_2$ obtained in July 2015 in the SCS to calculate a $\triangle pCO_2$, directly showing the difference of $pCO_2$ between surface water and atmosphere. We have also cited related paper to clarify that the negative $\triangle pCO_2$ in Transect 3 represented a strong $CO_2$ sink (Li et al., 2020). It's arbitrary to say "biological $CO_2$ sink", because this strong sink might not be totally caused by biological process. Thus we delete "biological". Residence time of $O_2$ at all CTD stations has been shown in Table 1 and 2.

**References:** Li, Q., Guo, X., Zhai, W., Xu, Y. and Dai, M.: Partial pressure of $CO_2$ and air-sea $CO_2$ fluxes in the South China Sea: Synthesis of an 18-year dataset, Prog. Oceanogr., 182, doi:10.1016/j.pocean.2020.102272, 2020.

Line 236 and elsewhere: Here and throughout the paper, you are reporting averages as ##.## +/- ##.##. You should specify that these represent avg +/- std dev: : : etc: : :
**Response:** Thanks for your suggestion. We have clarified this in the first paragraph of Results and Discussion:
*"In addition, please note that all averages we have published in this paper are reported in the format of mean ± standard deviation".*

Lines 299-301: Keep it clear in this sentence that upwelling does not necessarily produce an underestimation of NCP. It is more accurate to say that subsurface waters may have different (either more positive or more negative) $O_2$/Ar signatures that could produce either an underestimation or an underestimation of NCP, as you explain a few sentences later in the context of subsurface $O_2$ maxima in oligotrophic regions.
**Response:** Thanks for your suggestion. We have revised the sentences as "*Vertical mixing is considered the largest source of error in $O_2$/Ar-based NCP estimates because the upwelled subsurface water with different $O_2$/Ar signatures can produce either an overestimation or an underestimation of NCP in the mixed layer (Cassar et al., 2014; Izett et al., 2018). Former researches usually ignored the underestimated negative NCP that caused by vertical mixing (Giesbrecht et al., 2012; Reuer et al., 2007; Stanley et al., 2010).*"

Lines 313-315: The cutoff for salinity to account for waters influenced by shelf water injection is justified, but the cutoff intended to exclude regions with upwelling seems somewhat arbitrary.
**Response:** Sorry for that. We have specified the criteria of upwelling as "*the regions with negative NCP and the regions with salinity higher than 33.5 and temperature lower than 30 °C in Transect 4*". There are lots of overlap of these two criteria, effectively removing the upwelling regions.

Line 330: Consideration of other explanations is needed in this section as described in the general comments.
**Response:** Thanks for your suggestion. We have reported the residence time of $O_2$ and related it to the history of shelf water intrusion based on daily satellite-chl a data in the revised manuscript. Please see our reply to comments 1 for more detail.

Lines 412-418: As mentioned in the general comments, is it even meaningful to analyze the influence of light limitation on NCP when the euphotic zone is 2-7 times the depth of the mixed layer in this region? Much more discussion is needed to back this light limitation idea. Cassar et al 2011 makes the point for instance that MLD is not the only factor affecting light availability.
**Response:** Thanks for your suggestion. Because of your comment and the comments of reviewer 2, we have noticed the limitation of our analysis between MLD and $NCP_{vol}$. Cassar et al. (2011) pointed out that the correlation analysis between MLD and NCP actually reflected an iron-light co-limitation. Thus we decide not to use MLD as an indicator of light availability. In the revised manuscript, we calculate an average surface PAR by integrating the daily satellite-PAR data over the residence time of $O_2$ at each selected station in October 2014. Then we use light attenuation coefficient to calculate an average PAR in the mixed layer to make correlation analysis with NCP. Table 4 was added accordingly to show the results.

Lines 426-428: "This result implied that there was no significant productivity below the mixed layer that was missed by underway sampling." As mentioned in the general comments, you need to address the contradiction between this and subsurface $O_2$ maxima and potential deep chlorophyll maxima.

**Response:** Sorry for this arbitrary conclusion. We have deleted it. Subsurface chlorophyll maximum layer (SCML) existed at all stations in both cruises, while the obvious subsurface $O_2$ maxima just existed at Transect 1 and 2 in June 2015. We have reported these results in section 3.2. Both SCML and subsurface $O_2$ maxima were below the MLD, indicating that they might not influence the mixed layer NCP a lot in the region without vertical mixing.

Figure 1, lines 681 - 683: please describe what the point and star symbols represent.

Are these the locations of CTD casts? The fact that the color scale in (a) and (b) represents bathymetry should also be stated as well. The cruise path in (a) is also difficult to interpret based on the arrows. This figure is critical for the interpretation of subsequent figures. Perhaps numbering of points on the cruise plan would convey the path of travel better.

**Response:** Thanks for your suggestion. Following your suggestion, we redraw this figure and added an explanation of the symbols and color scales to the caption. Two insets have been added in Figure 1a and 1b to show the station numbers better.

[Figure]

**Figure 1.** Cruise tracks of two cruises in the slope region of the Northern South China Sea in **(a)** October 2014, **(b)** June 2015. The sea level height anomaly (SLA) and geostrophic current during observations in June 2015 (Chen et al., 2016) are shown in **(c)**. The black dots/stars represent the locations of the CTD casts. Red numbers indicate transects, while black numbers indicate the serial number of CTD stations based on the cruise plan. The color scale in **(a)** and **(b)** represents bathymetry.

**Technical corrections:**

Lines 310-318: Here and elsewhere throughout the manuscript, you use sentence structures that rely too heavily on parentheses, which can cause confusion for readers.These sentences in particular are very difficult to interpret. General: There are some grammatical errors in the conclusion

**Response:** We have reduced the number of parentheses in the text and revised the grammatical errors in the conclusion.

**Reply to Referee #2**

Thank you very much for your constructive comments. In the following we answer to your comments point by point and indicate how the manuscript is going to be revised.

**General comments**

In their manuscript "High-resolution distributions of $O_2/Ar$ on the northern slope of the South China Sea and estimates of net community production", the authors report continuous net community production (NCP) estimates in the mixed layer of the northern South China Sea (SCS). The study makes a clear contribution to understanding of productivity in marginal seas like the SCS, where prior NCP estimates are limited. To a lesser extent, the study also advances a relatively novel method to estimating NCP through continuous observations of $\Delta O_2/Ar$. My major critique is that the authors do not connect back to these original objectives in their paper. What new information have they gathered about the SCS as a result of this continuous method of measuring NCP, and how does this relate to past measurements of NCP in this region? Which methodological and/or environmental factors cause their estimates to compare or differ from past estimates? It is clear why their study is significant, but explicitly tying the discussion of results to these objectives will strengthen the scientific contributions of this paper.

  **Response:** Following your suggestion, we amended the manuscript and described the importance of shelf water intrusion in the study region from line 84 to 88 that *"The northern slope of SCS is an important transition region between coastal area and SCS basin. In the summer, the shelf water intrusion is an important process changing the nutritive state in the northern slope region of SCS (He et al., 2016; Lee Chen and Chen, 2006). But so far, the NCP enhancement caused by this process is still unknown."* Besides to figure out the regulating factor on NCP, we also set quantifying the contribution of shelf water to the NCP enhancement in the study region as one of our main objectives. This can be a new finding achieved by our $O_2/Ar$ method.

  By comparing our NCP result with previous results, we emphasized the important influence of shelf water intrusion in the summer. In addition, we highlighted that our high-resolution observation can catch the rapid NCP variation more effectively than previous methods. Related content can be found in section 3.3 and 3.4 (lines 346 to 354, lines 409 to 417).

  In the revised conclusion section, we pointed out that nitrogen is the main regulating factor on NCP in the study region and reported that *"the summer shelf water intrusion may significantly promote NCP by 376 %"*, connecting back to our objectives.

**Specific comments**

In the title: It is more accurate to write $\Delta O_2/Ar$ rather than $O_2/Ar$?
**Response:** We have changed $O_2/Ar$ to $\Delta(O_2/Ar)$

Line 35: Clarify what "indicator" means in this context, and in which conditions this assumption holds true (e.g., NCP may be partitioned into DOC production, particle export, zooplankton grazing, etc.).
**Response:** NCP here represents the net organic carbon production in the mixed layer, corresponding to the difference of phytoplankton photosynthesis and respiration. Thus we can use the mass balance of biological $O_2$ to quantify NCP. NCP can be regarded as a sum of biological POC, DOC and the organic carbon involved in the food wed. $O_2/Ar$-based NCP used to rely on the steady state assumption that the productive rate and wind speed keep constant. We have incorporated these into the revised manuscript from line 39 to 44 that *"Net community production (NCP) corresponds to gross primary production (GPP) minus community respiration (CR) in the water (Lockwood et al.,*

*2012; Stanley et al., 2010) and is an important indicator of carbon export. At steady state, NCP is equivalent to the rate of organic carbon export and transfer up the food web, which can quantify the strength of biological pump (Lockwood et al., 2012).”*

But recent researches pointed out that the $O_2$/Ar-based NCP estimate could be a time-weighted NCP over the residence time of $O_2$, weakening the need for the steady state assumption. We have clarified this in section 2.2.

Line 55: The water classifications are a bit confusing here. Perhaps it would be clearer to say that SCS water is a mix between two end-members: freshwater runoff from rivers and North Pacific offshore water.

**Response:** Sorry for the confusion caused. The SCS water is not just a mix between freshwater runoff and North Pacific offshore water. Because of its long residence time in the SCS region, its property has been changed a lot by heat exchange, precipitation and mixing processes. Li et al. (2018) regarded the SCS water as one of the end-members of water masses on the northern slope region of SCS. We have revised the manuscript to make this classification clearer. Now this content reads *“The surface water masses on the northern slope of SCS can be categorized into three regimes: shelf water, offshore water (e.g., the intruded Kuroshio water), and the SCS water (Feng, 1999; Li et al., 2018). The shelf water is mixed with fresh water from rivers or coastal currents and thus usually has low salinity (S < 33) and low density (Uu and Brankart, 1997; Su and Yuan, 2005; Cheng et al., 2014). Both offshore water and SCS water originate from the Northern Pacific. Thus offshore water has similar hydrographic characteristics of high temperature and high salinity as the Northern Pacific water. But the SCS water has changed a lot in its hydrographic property because of the mixing processes, heat exchange and precipitation during its long residence time in the SCS (Feng et al., 1999; Li et al., 2018).”*

**Reference:** Li, D., Zhou, M., Zhang, Z., Zhong, Y., Zhu, Y., Yang, C., Xu, M., Xu, D. and Hu, Z.: Intrusions of Kuroshio and Shelf Waters on Northern Slope of South China Sea in Summer 2015, J. Ocean Univ. China, 17(3), 477–486, doi:10.1007/s11802-018-3384-2, 2018.

Lines 70-84: It is unclear what the aim of listing these numbers is? Do the authors wish to convey that NCP is variable across SCS studies? It would be useful to reference these numbers again in the discussion for comparison. In any case, when reporting NCP and export, use both $O_2$ and C units so that the numbers are comparable. The authors can perhaps apply the photosynthetic quotient used in the method to do this conversion to keep units consistent (line 174).

Line 79: Should the units here be s^-1 rather than a^-1?

**Response:** Here we want to report the previous researches about carbon export in the SCS. The POC just occupies a portion of NCP, and its value may not be very comparable with NCP. Hence we deleted the description about previous POC export and mainly focus on the previous NCP researches in the SCS in this paragraph. We have cited the previous NCP values to compare with our NCP results in this study. We have also converted the unit of all NCP values mentioned in the text to “mmol C $m^{-2}$ $d^{-1}$”.

Lines 85-86: Describe the potential inaccuracies of each discrete method so that it is clearer how this study benefits scientific understanding of the SCS.

**Response:** Thanks for your suggestion. We revised the manuscript to describe the potential inaccuracies and shortcomings of discrete methods in the following paragraph as that *“Discrete sampling suffers from low spatial resolution, and cannot adequately resolve variabilities caused by small-scale physical or biological processes in dynamic marine systems. In addition, each of the three methods for NCP estimate mentioned above has its limitation. DIC-based NCP estimate is not suitable for the coastal region, because instead of biological*

*metabolism, the terrestrial runoff can be the strongest factor influencing the DIC in the coastal system (Mathis et al., 2011). The inavoidable difference between in situ circumstance and on deck incubation condition can introduce uncertainties to the NCP derived from light/dark bottle incubation (Grande et al., 1989). Though Argo profiling float partly gets rid of the limitation of discrete sampling, it's hard to control its movement in the study region. However, no high-resolution measurement of NCP has been reported for the SCS so far."*

Section 2.2: Explain how the 5-minute NCP values are scaled up to daily estimates.

Lines 179-183: As written, these two sentences imply that the authors do not know whether their NCP estimates represent daily or monthly signals. If they represent the latter, would this not defeat the purpose of the study, which is to resolve "highly dynamic environmental fluctuations of coastal systems" (line 87) in shorter than monthly time scales?

**Response:** The "5-min" here is not a timescale for monitoring the net change of a biological production tracer during this period to calculate the daily NCP, but the time interval of our underway data. The 5-min interval is usually along with a spatial interval of 500 m to 1 km because the ship is moving, thus it can also be regarded as the spatial resolution of underway sampling, which is much higher than that of discrete sampling (e.g., CTD cast).

The "over the past month" used in the previous version is not very accurate, and we have revised it to "*during the residence time of oxygen in the mixed layer*". The $\Delta(O_2/Ar)$ we obtained is a cumulative result that had been influenced by the physical (e.g., air-sea exchange, water mixing) and biological processes (e.g., respiration and photosynthesis) during the residence time of oxygen in the mixed layer. The influence of "environmental fluctuations" during that period could be reflected by the physical and biological processes mentioned above, which certainly had a contribution to the final NCP values we estimated. That's why we can catch the dramatic high NCP and negative NCP resulted from shelf water intrusion and upwelling respectively in the June cruise.

Here we have intended to clarify that our NCP estimate is a time-weighted result instead of a daily average result. If the environment is at the steady state (e.g., constant productive rate and constant wind), our estimate can be an actual daily NCP. But in reality, steady state is always violated because of the variable wind-speed (or gas transfer velocity) over time. We apply a time-weighted scheme to calculate the gas transfer velocity following Reuer et al.(2007) and Teeter et al.(2018), more heavily weighting recent periods and storm periods to erase the importance of earlier states. As a result, though our NCP result is in the unit of *mmol C m$^{-2}$ d$^{-1}$*, it's not the actual daily NCP but represents an estimate of time-weighted sea-to-air biological oxygen flux over the residence time of oxygen before our observation of $O_2/Ar$.

**Reference:** Reuer, M. K., Barnett, B. A., Bender, M. L., Falkowski, P. G. and Hendricks, M. B.: New estimates of Southern Ocean biological production rates from O$_2$/Ar ratios and the triple isotope composition of O$_2$, Deep Sea Res. Part I Oceanogr. Res. Pap., 54(6), 951–974, doi:10.1016/j.dsr.2007.02.007, 2007.

Teeter, L., Hamme, R. C., Ianson, D. and Bianucci, L.: Accurate Estimation of Net Community Production From O$_2$/Ar Measurements, Global Biogeochem. Cycles, 32(8), 1163–1181, doi:10.1029/2017GB005874, 2018.

Lines 412-416: Clarify how the DIN and NCP criteria were chosen for each cruise.

Fig. 10b is not very compelling as the lowest MLD - highest volumetric NCP data point seems to drive the negative correlation. Thus, the authors should consider removing their analysis of June 2015 data from Fig. 10b, and just discuss the analysis in the text in relation to the much stronger relationship between MLD and volumetric NCP during October 2014. Another related analysis that may be interesting is comparing NCP values at stations where MLD is deeper than the euphotic zone depth, to NCP at stations where the MLD is shallower than the euphotic zone depth.

**Response:** Thanks for your suggestion. The relationship shown in figure 10b is not convincing enough because of inadequate data points. In addition, during the June 2015 cruise, the euphotic zone was 2-7 times the depth of the mixed layer, thus it's not meaningful to discuss the light limitation in the summer. We selected 9 stations of October cruise where surface DIN concentration in the range of 0.10─0.17 μmol L$^{-1}$ to make the analysis of NCP and light. Because of your comment and the comments of reviewer 1, we have noticed the limitation of our analysis between MLD and $NCP_{vol}$. The negative correlation between $NCP_{vol}$ and MLD we obtained may partly result from that $NCP_{vol}$ is calculated by NCP/MLD. Thus we calculate an average surface PAR by integrating the daily satellite-PAR data obtained from NASA ocean color website (https://oceancolor.gsfc.nasa.gov/l3) over the residence time of $O_2$ at each selected station in October 2014. 8-day PAR data were used to estimate the missing daily data. Then we use light attenuation coefficient ($K_d$) to calculate an average PAR in the mixed layer to make a correlation analysis with NCP. The results were shown in a new table (Table 4). This new analysis gives a result that light availability is not a limitation on NCP in the SCS, much more convincing than the former analysis just based on MLD.

The calculation of $K_d$ basically based on Lambert-Beer law (Kirk 1994; Jerlov 1976):

$$K_d = -\frac{1}{z} \ln \frac{E_d(z)}{E_d(0)} = \frac{4.605}{Z_{eu}}$$

Where $K_d$ (m$^{-1}$) is the light attenuation coefficient in the euphotic layer; $E_d(0)$ is the PAR at the surface, integrating an average over the residence time of $O_2$ before our observation, in the unit of mol m$^{-2}$ d$^{-1}$; z represents a depth (m) and $E_d(z)$ is the PAR at this depth; $Z_{eu}$ is the euphotic depth (m).

**Table 4.** Satellite-PAR data and NCP at selected stations in October 2014

| Station | Date of observation | MLD (m) | $Z_{eu}$ (m) | Surface PAR[a] (mol m$^{-2}$ d$^{-1}$) | $K_d$ (m$^{-1}$) | ML PAR[b] (mol m$^{-2}$ d$^{-1}$) | NCP (mmol C m$^{-2}$ d$^{-1}$) |
|---------|---------------------|---------|--------------|----------------------------------------|------------------|-----------------------------------|--------------------------------|
| O-01 | 2014/10/13 | 58 | 82 | 42.0 | $5.6 * 10^{-2}$ | 12.0 | 3.0 |
| O-02 | 2014/10/13 | 64 | 74 | 42.0 | $6.2 * 10^{-2}$ | 10.0 | 15.1 |
| O-03 | 2014/10/14 | 56 | 84 | 41.1 | $5.5 * 10^{-2}$ | 12.4 | 10.1 |
| O-08 | 2014/10/21 | 49 | 72 | 38.7 | $6.4 * 10^{-2}$ | 11.4 | 15.7 |
| O-10 | 2014/10/15 | 68 | 81 | 40.0 | $5.7 * 10^{-2}$ | 9.8 | 4.4 |
| O-13 | 2014/10/16 | 48 | 52 | 39.2 | $8.9 * 10^{-2}$ | 8.7 | 15.3 |
| O-15 | 2014/10/22 | 49 | 68 | 38.6 | $6.8 * 10^{-2}$ | 10.8 | 16.3 |
| O-20 | 2014/10/18 | 35 | 61 | 39.2 | $7.5 * 10^{-2}$ | 13.3 | 16.4 |
| O-22 | 2014/10/17 | 76 | 102 | 42.2 | $4.5 * 10^{-2}$ | 11.6 | 15.7 |

[a] Average surface PAR over the residence time of $O_2$ in the mixed layer. [b] Average PAR in the mixed layer.

**References:** Kirk, J. T.: Light and photosynthesis in aquatic ecosystems, Cambridge university press, UK, 1994.

Jerlov, N. G.: Marine optics, Elsevier, Netherlands, 1976.

**Technical comments**

Figure 1: Explain what the dots/markers in the panels represent. Are they the locations of the CTD casts? If not, it is worth adding the locations of the CTD casts to this figure so that readers may better understand the interpolation of MLD between casts for underway data.

**Response:** Yes, the dots and stars are the locations of the CTD casts. Following your suggestion, we have added an explanation of these markers to the caption of figure 1, "*the black dots/stars represent the locations of the CTD casts*".

Figure 3: Why were there more variables in the June cruise? This is not clear in the methods.
**Response:** Sorry for that. During the cruise in October 2014, the DO sensor of RBR broke down, and we did not make the standards for $CO_2$ calibration. Thus there are no data of DO and $pCO_2$ in that cruise. We have clarified this in the section 2.1 which reads:
*"We didn't obtain continuous DO data in October 2014 because the DO sensor of RBR broke down during this cruise."*
*"The instrumental $CO_2$ ion current was calibrated at about 12–24 h intervals using equilibrated seawater standards as per Guéguen and Tortell (2008) during the survey in June 2015."*

Figure 5: Write in the salinity units. It is worth clarifying somewhere in the figure text, as well as the main text referencing Fig. 5, that the temperature fluctuations shown here are too small to reflect upwelling.
Figure 6: Write in the salinity units.
Figure 8: Write in the salinity units.
Figure 9: Write in the salinity units.
**Response:** Thanks for your suggestions. We have added the units of salinity on these figures as well as figure 2 & 3. We have also clarified in the main text that the temperature fluctuations shown in figure 5 are too small to reflect upwelling.

Figure S1: This actually is referenced after Fig. S2 so consider switching the figure order. Why is [$NO_3^-$] omitted here?
**Response:** Thanks for your suggestion. We have switched the order of figure S1 & S2. The surface $NO_3^-$ concentration was below the detection limit at all sampling stations during the cruise in 2014, thus we didn't make the plot for [$NO_3^-$]. We had clarified this in the figure caption that *"The surface concentration of nitrate ($NO_3^-$) at all sampling stations was below the detection limit during this cruise."*.

Figures S3-S7: These are not referenced in the text, but they should be if they are to be published. Otherwise, it is not clear what the significance of showing these data are, as they could just go on an online repository which gets referenced in the text.
**Response:** Thanks for your suggestion. These transects are not very representative, so we didn't discuss them in the text. We decided to delete these figures from the supplementary. But the data of these transects can be easily downloaded from the online repository we shared.

[revised manuscript text omitted]

**Supplementary**

**Table S1a.** Correlation coefficient matrix of PCA in October 2014

| | | Temperature (T) | Salinity (S) | DIN | DIP | DSi | MLD | Chl a | $\Delta(O_2/Ar)$ | NCP |
|---|---|---|---|---|---|---|---|---|---|---|
| Correlation coefficient | Temperature (T) | 1.000 | 0.190 | −0.536 | −0.305 | 0.539 | 0.267 | −0.162 | −0.524 | −0.757 |
| | Salinity (S) | 0.190 | 1.000 | 0.098 | −0.337 | −0.492 | 0.352 | 0.145 | −0.295 | −0.359 |
| | DIN | −0.536 | 0.098 | 1.000 | 0.000 | −0.132 | 0.176 | 0.125 | 0.574 | 0.706 |
| | DIP | −0.305 | −0.337 | 0.000 | 1.000 | −0.264 | −0.494 | −0.267 | −0.075 | 0.135 |
| | DSi | 0.539 | −0.492 | −0.132 | −0.264 | 1.000 | 0.098 | 0.051 | −0.024 | −0.171 |
| | MLD | 0.267 | 0.352 | 0.176 | −0.494 | 0.098 | 1.000 | 0.068 | 0.202 | −0.037 |
| | Chl a | −0.162 | 0.145 | 0.125 | −0.267 | 0.051 | 0.068 | 1.000 | 0.397 | 0.323 |
| | $\Delta(O_2/Ar)$ | −0.524 | −0.295 | 0.574 | −0.075 | −0.024 | 0.202 | 0.397 | 1.000 | 0.906 |
| | NCP | −0.757 | −0.359 | 0.706 | 0.135 | −0.171 | −0.037 | 0.323 | 0.906 | 1.000 |
| Statistical significance | Temperature (T) | | 0.288 | 0.044 | 0.181 | 0.043 | 0.214 | 0.317 | 0.049 | 0.003 |
| | Salinity (S) | 0.288 | | 0.388 | 0.155 | 0.062 | 0.144 | 0.335 | 0.189 | 0.139 |
| | DIN | 0.044 | 0.388 | | 0.500 | 0.350 | 0.302 | 0.358 | 0.032 | 0.008 |
| | DIP | 0.181 | 0.155 | 0.500 | | 0.216 | 0.061 | 0.214 | 0.413 | 0.346 |
| | DSi | 0.043 | 0.062 | 0.350 | 0.216 | | 0.387 | 0.441 | 0.473 | 0.307 |
| | MLD | 0.214 | 0.144 | 0.302 | 0.061 | 0.387 | | 0.422 | 0.275 | 0.457 |
| | Chl a | 0.317 | 0.335 | 0.358 | 0.214 | 0.441 | 0.422 | | 0.113 | 0.167 |
| | $\Delta(O_2/Ar)$ | 0.049 | 0.189 | 0.032 | 0.413 | 0.473 | 0.275 | 0.113 | | 0.000 |
| | NCP | 0.003 | 0.139 | 0.008 | 0.346 | 0.307 | 0.457 | 0.167 | 0.000 | |

**Table S1b.** Component matrix of variables in October 2014

|  | Factor 1 | Factor 2 |
|---|---|---|
| Temperature (T) | −0.847 | 0.264 |
| Salinity (S) | −0.259 | 0.521 |
| DIN | 0.741 | 0.259 |
| DIP | 0.189 | −0.817 |
| DSi | −0.288 | 0.156 |
| MLD | −0.065 | 0.793 |
| Chl a | 0.343 | 0.456 |
| $\Delta(O_2/Ar)$ | 0.858 | 0.276 |
| NCP | 0.979 | 0.026 |

**Table S2a.** Correlation coefficient matrix of PCA in June 2015

| | | Temperature (T) | Salinity (S) | DIN | DIP | DSi | MLD | Chl a | $pCO_2$ | DO | $\Delta(O_2/Ar)$ | NCP |
|---|---|---|---|---|---|---|---|---|---|---|---|---|
| Correlation coefficient | Temperature (T) | 1.000 | −0.128 | 0.217 | 0.150 | −0.239 | −0.244 | −0.156 | −0.189 | −0.313 | 0.060 | −0.224 |
| | Salinity (S) | −0.128 | 1.000 | −0.873 | 0.163 | 0.301 | 0.614 | −0.921 | 0.859 | −0.831 | −0.816 | −0.787 |
| | DIN | 0.217 | −0.873 | 1.000 | −0.067 | −0.260 | −0.594 | 0.754 | −0.705 | 0.736 | 0.910 | 0.747 |
| | DIP | 0.150 | 0.163 | −0.067 | 1.000 | −0.222 | −0.017 | −0.349 | 0.165 | −0.355 | −0.172 | −0.195 |
| | DSi | −0.239 | 0.301 | −0.260 | −0.222 | 1.000 | 0.474 | −0.275 | 0.443 | −0.241 | −0.361 | −0.276 |
| | MLD | −0.244 | 0.614 | −0.594 | −0.017 | 0.474 | 1.000 | −0.593 | 0.816 | −0.541 | −0.507 | −0.518 |
| | Chl a | −0.156 | −0.912 | 0.754 | −0.349 | −0.275 | −0.593 | 1.000 | −0.867 | 0.948 | 0.793 | 0.884 |
| | $pCO_2$ | −0.189 | 0.859 | −0.705 | 0.165 | 0.443 | 0.816 | −0.867 | 1.000 | −0.762 | −0.701 | −0.767 |
| | DO | −0.313 | −0.831 | 0.736 | −0.355 | −0.241 | −0.541 | 0.948 | −0.762 | 1.000 | 0.839 | 0.946 |
| | $\Delta(O_2/Ar)$ | 0.060 | −0.816 | 0.910 | −0.172 | −0.361 | −0.507 | 0.793 | −0.701 | 0.839 | 1.000 | 0.846 |
| | NCP | −0.224 | −0.787 | 0.747 | −0.195 | −0.276 | −0.518 | 0.884 | −0.767 | 0.946 | 0.846 | 1.000 |
| Statistical significance | Temperature (T) | | 0.331 | 0.228 | 0.305 | 0.205 | 0.200 | 0.297 | 0.259 | 0.138 | 0.420 | 0.220 |
| | Salinity (S) | 0.331 | | 0.000 | 0.288 | 0.148 | 0.010 | 0.000 | 0.000 | 0.000 | 0.000 | 0.000 |
| | DIN | 0.228 | 0.000 | | 0.410 | 0.185 | 0.013 | 0.001 | 0.002 | 0.001 | 0.000 | 0.001 |
| | DIP | 0.305 | 0.288 | 0.410 | | 0.223 | 0.477 | 0.111 | 0.286 | 0.106 | 0.279 | 0.252 |
| | DSi | 0.205 | 0.148 | 0.185 | 0.223 | | 0.043 | 0.170 | 0.056 | 0.203 | 0.102 | 0.170 |
| | MLD | 0.200 | 0.010 | 0.013 | 0.477 | 0.043 | | 0.013 | 0.000 | 0.023 | 0.032 | 0.029 |
| | Chl a | 0.297 | 0.000 | 0.001 | 0.111 | 0.170 | 0.013 | | 0.000 | 0.000 | 0.000 | 0.000 |
| | $pCO_2$ | 0.259 | 0.000 | 0.002 | 0.286 | 0.056 | 0.000 | 0.000 | | 0.001 | 0.003 | 0.001 |
| | DO | 0.138 | 0.000 | 0.001 | 0.106 | 0.203 | 0.023 | 0.000 | 0.001 | | 0.000 | 0.000 |
| | $\Delta(O_2/Ar)$ | 0.420 | 0.000 | 0.000 | 0.279 | 0.102 | 0.032 | 0.000 | 0.003 | 0.000 | | 0.000 |
| | NCP | 0.220 | 0.000 | 0.001 | 0.252 | 0.170 | 0.029 | 0.000 | 0.001 | 0.000 | 0.000 | |

**Table S2b.** Component matrix of variables in June 2015

|  | Factor 1 | Factor 2 |
|---|---|---|
| Temperature (T) | 0.024 | −0.786 |
| Salinity (S) | −0.936 | 0.043 |
| DIN | 0.876 | −0.132 |
| DIP | −0.223 | −0.601 |
| DSi | −0.405 | 0.582 |
| MLD | −0.718 | 0.402 |
| Chl a | 0.950 | 0.217 |
| $p\text{CO}_2$ | −0.908 | 0.186 |
| DO | 0.927 | 0.340 |
| $\Delta(\text{O}_2/\text{Ar})$ | 0.902 | 0.008 |
| NCP | 0.909 | 0.227 |

[Figure]

**Figure S1.** Surface distributions of **(a)** nitrate (NO₃⁻), **(b)** nitrite (NO₂⁻), **(c)** ammonium (NH₄⁺), **(d)** phosphate (PO₄³⁻), **(e)** silicate (SiO₃²⁻) and **(f)** mixed layer depth (MLD) in June 2015. We regarded the nutrients data that were below the detection limit as "0" when made these plots.

[Figure]

**Figure S2.** Surface distributions of **(a)** nitrite ($NO_2^-$), **(b)** ammonium ($NH_4^+$), **(c)** phosphate ($PO_4^{3-}$), **(d)** silicate ($SiO_3^{2-}$)
and **(e)** mixed layer depth (MLD) in October 2014. The surface concentration of nitrate ($NO_3^-$) at all sampling stations was
below the detection limit during this cruise. We regarded the nutrients data that were below the detection limit as "0" when
made these plots.

[Figure]

**Figure S3.** Daily satellite-chlorophyll on selected date from June 10 to 25, 2015. Stars represent CTD locations. We roughly set light blue (represents ~ 0.2 μg L$^{-1}$) in this figure as the criterion of shelf water. This figure was made based on the M_Map mapping package for MATLAB (Pawlowicz, R., 2020. "M_Map: A mapping package for MATLAB", version 1.4m, [Computer software], available online at www.eoas.ubc.ca/~rich/map.html.).

---

## Referee Report (RR1)

General comments:

Qin et al have thoughtfully revisited their manuscript and made a number of revisions that have strengthened the paper overall. In particular, their analysis of the timing of shelf water intrusion into the SCS versus the residence time of their O2/Ar measurements, supported with satellite chlorophyll data, provides convincing and valuable support of their conclusion regarding the contribution of shelf water intrusion to NCP.

The authors' more detailed analysis of average mixed-layer PAR and their revised conclusion that light does not limit mixed-layer NCP in the study region is also an important improvement. Generally, the revisions have demonstrated care and critical thought in reevaluating the interpretation of this study's findings. The changes made have satisfied this reviewer's original criticisms of the manuscript.

The new assessment of the impact of the shelf water intrusion upon observed NCP rates is also quite clever and an interesting scientific contribution.

Specific comments:

Figure S3 is quite nice and I'm very tempted to recommend that this be included as a main figure. I certainly think it adds more value to the main article than Figure 9, for instance.

Line 15: The statement that NCP is a proxy of carbon export is slightly too strong, as NCP is more accurately a metric of export potential (excess organic matter production available for export to depth).

Section 2.3: How many replicates for nutrient analysis were collected at each CTD station?

Line 376-378: Upon further reflection, this statement reads as attributing somewhat too strong of a causal relationship. I find the NH4 measurements, sparse though they are, to be useful evidence of ammonium contributing to the peak in NCP on this transect, and the residence time at these stations was quite short which further supports this, but at the end of the day these are just two stations. This also further emphasizes the importance of replication of nutrient sampling as noted above. If these are only single measurements, then only a very weak statement can be made here, and the associated discussion should be reconsidered more thoroughly.

Figure 5c  and Figure 6c: As noted above, if the NH4 nutrient sampling includes multiple measurements, the individual replicates in addition to the mean might be shown.

Lines 385-387: I would cite Figure S3 here, as this clearly shows the influence of shelf water.

Line 421 (and line 27 in abstract as well as line 559 in the conclusion): The figure of 376% is a little overly precise, especially given the variance in the NCP of the background and intrusion-influenced water masses. I would replace with a more general statement like "by

potentially more than threefold" or similar, following the convention the authors have adopted in lines 492-495.

Lines 510-529: this new passage and the associated new data figure and table are very strong additions. Again would make the case for Figure S3 to become a main figure given its importance to the manuscript's conclusions regarding the July cruise.

Throughout: I would recommend that the authors double-check the manuscript text for minor grammatical errors, particularly in the newly-added text.

For instance, in "Dissolved oxygen-to-argon ratios (O2/Ar) in the oceanic mixed layer has been widely used" (Line 13), "has" should be replaced with "have".

Similarly, "Despite the coastal waters such as shelves and estuaries only account for 7 % of the global ocean surface area" (line 52) should be revised to something like "Despite coastal waters such as shelves and estuaries only accounting for 7 % of the global ocean surface area", etc.

---

## Author Response (AR2)

**Response to editor:**

Topic Editor Decision: Publish subject to minor revisions (review by editor) (08 Dec 2020) by Mario Hoppema Comments to the Author: Dear Dr. Zhang and co-authors,

Thank you for submission of your revised manuscript. Referee #1 is generally satisfied with your revisions, though some minor issues remain. The referee described those in the attached referee report.

I agree with the referee and think the manuscript is almost there. I did go through the manuscript and provide minor and technical comments below. Please account for the comments by the referee and me, and submit your final version of the manuscript

**Response:** Thank you very much for your constructive comments. We have considered all suggestions and incorporated them into the revised version. In the following we answer to your comments point by point and indicate how the manuscript is going to be revised.

You are using data from other sources, e.g. satellite data from Copernicus and NASA. Please check and confirm whether you fulfill the fair data use statement and whether the acknowledgment of the data use is according to the request by the data providers.

**Response:** Thanks for this suggestion. We have added related statement to the Acknowledgement and citation to the reference list following the requests of Copernicus and NASA.

L42-44 "NCP is equivalent to the rate of organic carbon export and transfer up the food web, which can quantify the strength of biological pump (Lockwood et al., 2012)." It is not clear to me what "transfer up the food web" means, as this is also compared and stands besides carbon export. After the comma, I suggest: and is a measure for the strength of the biological pump (Lockwood et al., 2012). **Response:** Sorry for the confusion we made. NCP is the difference between gross primary production and respiration, and can also represent net organic carbon production. At steady state, organic carbon mainly has two fates: being directly exported to the deep water and involved into the food web. That's why we write this sentence. But after reconsidering, we think the organic carbon that's involved into the food web will finally be exported to the deep water through the sinking of fecal pellets and dead creatures. Thus we decide to delete "transfer up the food web".

L44-45 "Dissolved oxygen-to-argon ratio  $(O_2/Ar)$  has been developed as a proxy for NCP ..." I think this requires some more explanation why this is the case.

**Response:** We agree with you. Prior to this sentence, we have added an explanation that "*NCP* effectively couples carbon cycle and oxygen ( $O_2$ ) production through photosynthesis and respiration in the euphotic layer, thus many previous researches measured the mass balance of  $O_2$  to quantify NCP (e.g., Emerson et al., 1991; Hendricks et al., 2004; Huang et al., 2012; Reuer et al., 2007). Argon (Ar), a biological inert gas, was commonly used to normalize the  $O_2$  concentration in these researches. Based on the similar solubility properties of  $O_2$  and Ar, oxygen-to-argon ratio ( $O_2/Ar$ ) can remove the influences of physical processes (i.e., temperature and pressure change, bubble injection) on the mass balance of  $O_2$  (Craig and Hayward, 1987)."

L56-57 "in global NCP data sets" It was not clear to me that these exist. Please add a reference. **Response:** Sorry for this inappropriate expression. The "global NCP data sets" don't exist. Here we want to emphasize that coastal and shelf regions still suffer from low resolution measurements. We have revised this sentence as "*However, these regions still suffer from low resolution measurements that can't provide representative high-resolution NCP data.*"

L59 "extremely complex ecological characteristics" It is not clear to me what that implies. Why is it more complex than other regions? Please explain or tone down. **Response:** We agree that "extremely" is exaggerated. We have deleted it.

L72 "during its long residence time in the SCS" Please be more specific; how long? **Response:** We have clarified in the text that the residence time can be about 40 years.

L127 If you measure concentrations, please indicate that here.

**Response:** Thanks for the suggestion. We have changed this sentence to be '*Continuous measurements* of dissolved gases ( $O_2$ , Ar, and  $CO_2$ ) were obtained using membrane inlet mass spectrometry'. But we think it's not appropriate to indicate concentration or partial pressure here. MIMS just provides the signal intensities of the dissolved gases instead of the actual concentrations. O2/Ar-standards were not available to calibrate the oxygen and argon signals from the MIMS. However, the O2 signal was normalized to Ar to yield a biologically relevant parameter, i.e. O2/Ar, which can be directly calculated using the signal intensities of O2 and Ar. Temperature-controlled seawater standards were used to calibrate  $pCO_2$  and biological O2 saturation ( $\Delta O_2$ /Ar) measurements. Signal intensity can be a reflection of the concentration or partial pressure of the dissolved gas; thus we can use a calibration curve to quantify  $pCO_2$ .

L152 Please indicate what is measured here of  $O_2$ , Ar and  $CO_2$ : concentration, partial pressure? **Response:** Due to the same reason as mentioned above, we mean the long-term signal stability for  $O_2$ , Ar and  $CO_2$  here, hence it's not necessary to indicate concentration or partial pressure.

L129-131 Are there any cruise reports that could be cited? Do the cruises have EXPO codes? **Response:** Sorry, there's no available report that could be cited here. And the cruises don't have EXPO codes.

L156 delete "down during this cruise" L264 delete "In addition," L295-296 Change to: ... of the two cruises are shown in Tables 1 and 2. L299 delete "reasonable" L303-304 delete "shallower than that of October 2014" L530 I suggest: The amount of light may also play a role in the extent of primary production. (that light play a role in primary production is trivial)

Response: Thanks for your suggestions. We have done these.

L357-358 Please explain why you took these transects. Why not take transect 4, which was sampled

in both years.

**Response:** There are several reasons. Both selected transects are uninterrupted, with relatively short time for CTD casts, which can give a good view of the zonal/ meridional variations of parameters. In addition, highest DIN and associated spike of NCP occurred in Transect 5 in 2014, thus we wanted to highlight this result. Transect 4 in 2015 was chosen to discuss the influence of upwelling, which is quite meaningful to refine our NCP results. Transect 4 in 2014 was partly interrupted by an obvious ship's drift during the period for CTD casts (Figure 2, 4) and the DIN concentration in this transect was not very significant, so we didn't take this transect.

Table 1 Please use data format like 13 Oct 2014

Figure 2 Please delete PSU at the salinity panel

Figure 3 Please delete PSU at the salinity panel

Figure 5a Please delete PSU at the salinity panel

Figure 6a Please delete PSU at the salinity panel

Figure 7 The labels at the axes are too small. Please make them larger for better readability.

Figure 8a Please delete PSU at the salinity panel

Figure 9b and 9d Please delete PSU at the salinity panel

**Response:** Thanks for your suggestions. We have done these. All the dates in Table 1, 2, 3 and 4 have been revised to the format like 13 Oct 2014.

**Response to reviewer 1:**

General comments:

Qin et al have thoughtfully revisited their manuscript and made a number of revisions that have strengthened the paper overall. In particular, their analysis of the timing of shelf water intrusion into the SCS versus the residence time of their  $O_2/Ar$  measurements, supported with satellite chlorophyll data, provides convincing and valuable support of their conclusion regarding the contribution of shelf water intrusion to NCP.

The authors' more detailed analysis of average mixed-layer PAR and their revised conclusion that light does not limit mixed-layer NCP in the study region is also an important improvement.

Generally, the revisions have demonstrated care and critical thought in reevaluating the interpretation of this study's findings. The changes made have satisfied this reviewer's original criticisms of the manuscript.

The new assessment of the impact of the shelf water intrusion upon observed NCP rates is also quite clever and an interesting scientific contribution.

**Response:** Thank you very much for your constructive comments. We have considered all suggestions and incorporated them into the revised version. In the following we answer to your comments point by point and indicate how the manuscript is going to be revised.

Specific comments:

Figure S3 is quite nice and I'm very tempted to recommend that this be included as a main figure. I certainly think it adds more value to the main article than Figure 9, for instance. **Response:** Thanks for your suggestion. We have set this figure as Figure 10.

Line 15: The statement that NCP is a proxy of carbon export is slightly too strong, as NCP is more accurately a metric of export potential (excess organic matter production available for export to depth). **Response:** Thanks for your suggestion. We have revised this sentence as "....*is a measure for the strength of the biological pump.*".

Section 2.3: How many replicates for nutrient analysis were collected at each CTD station?

Line 376-378: Upon further reflection, this statement reads as attributing somewhat too strong of a causal relationship. I find the NH4 measurements, sparse though they are, to be useful evidence of ammonium contributing to the peak in NCP on this transect, and the residence time at these stations was quite short which further supports this, but at the end of the day these are just two stations. This also further emphasizes the importance of replication of nutrient sampling as noted above. If these are only single measurements, then only a very weak statement can be made here, and the associated discussion should be reconsidered more thoroughly.

Figure 5c and Figure 6c: As noted above, if the NH4 nutrient sampling includes multiple measurements, the individual replicates in addition to the mean might be shown.

**Response:** We only made single measurement for the nutrients. Our method has a good precision of 3 %, ensuring the reliability of the single nutrients data. Therefore, we didn't make multiple sampling at each CTD station. We agree with your concern that the statement here is too strong, since there were only two data points of NH4 in this transect. So we have toned down the statement that "*However, we only got nutrient data at two CTD stations in this transect, thus the result we obtained here just*

**indicated that high NCP occurred at the station with relatively high $NH_4^+$ , but couldn't be a strong evidence that $NH_4^+$ was the main factor influencing NCP in this transect."**

Lines 385-387: I would cite Figure S3 here, as this clearly shows the influence of shelf water. **Response:** Thanks for your suggestion. But we think after we set S3 as a main figure (Figure 10), citing it here may make the order of the main figures a bit chaotic. Because if we cite it here, it should be "Figure 7" based on the order of being cited. But the paragraph that discusses this figure is near the end of the *Results and Discussion* section where "Figure 10" is much more appropriate.

Line 421 (and line 27 in abstract as well as line 559 in the conclusion): The figure of 376% is a little overly precise, especially given the variance in the NCP of the background and intrusion-influenced water masses. I would replace with a more general statement like "by potentially more than threefold" or similar, following the convention the authors have adopted in lines 492-495.

**Response:** Thanks for your suggestion. The variance in the NCP of the background and intrusioninfluenced water should be considered. We have replaced "376 %" with "potentially more than threefold" in these sentences you mentioned.

Lines 510-529: this new passage and the associated new data figure and table are very strong additions. Again would make the case for Figure S3 to become a main figure given its importance to the manuscript's conclusions regarding the July cruise.

Response: Thanks for your suggestion. We have set Figure S3 as Figure 10 in the main text.

Throughout: I would recommend that the authors double-check the manuscript text for minor grammatical errors, particularly in the newly-added text.

For instance, in "Dissolved oxygen-to-argon ratios  $(O_2/Ar)$  in the oceanic mixed layer has been widely used" (Line 13), "has" should be replaced with "have".

Similarly, "Despite the coastal waters such as shelves and estuaries only account for 7 % of the global ocean surface area" (line 52) should be revised to something like "Despite coastal waters such as shelves and estuaries only accounting for 7 % of the global ocean surface area", etc.

**Response:** Thanks for your suggestion. We have revised the grammatical errors.

---

## Author Response (AR3)

Dear Dr. Hoppema,

Thank you very much for helping improve our manuscript and accepting it.

When I uploaded files, I noticed that there was a term named "Author's certification" in the submission system, which said that "I herewith certify that the content (e.g. text, figures, equations, or tables) of all files uploaded in this form is exactly the same as the version of my manuscript accepted by the Topic Editor." But here we want to clarify that we actually made some slight changes in figures and the main text following the submission guideline.
 (https://www.ocean-science.net/for_authors/submit_your_manuscript.html).

Specific changes:
**Figure 1:** We added legends which clarified all symbols in the figure.
**Figure 10:** We added a legend to clarify the star symbols in this figure. And we also changed the format and font of the longitude and latitude labels to make them consistent with other figures.
**In the main text:** We added a space between degree sign and the direction to the coordinates (e.g. 19.7° N, 115.6° E). And we numbered the equations with Arabic numerals in parentheses on their right-hand side.

We didn't make any changes in the main content of figures and the main text.

Best wishes
Chuan Qin